# Influence of atmospheric in-cloud aqueous-phase chemistry on global simulation of $SO_2$ in CESM2

Wendong Ge[1], Junfeng Liu[1], Kan Yi[2], Jiayu Xu[1], Yizhou Zhang[1], Xiurong Hu[3], Jianmin Ma[1], Xuejun Wang[1], Yi Wan[1], Jianying Hu[1], Zhaobin Zhang[1], Xilong Wang[1], Shu Tao[1]

[1]Laboratory for Earth Surface Processes, College of Urban and Environmental Sciences, Peking University, Beijing, 100871, China

[2]Institute of Science and Technology, China Three Gorges Corporation, Beijing, 100038, China

[3]College of Economics and Management, Nanjing University of Aeronautics and Astronautics, Nanjing, 211106, China

*Correspondence to*: Junfeng Liu (jfliu@pku.edu.cn)

**Abstract.** Sulfur dioxide ($SO_2$) is a major atmospheric pollutant and precursor of sulfate aerosols, which influences air quality, cloud microphysics and climate. Therefore, better understanding the conversion of $SO_2$ to sulfate is essential to simulate and predict sulfur compounds more accurately. This study evaluates the effects of in-cloud aqueous-phase chemistry on $SO_2$ oxidation in the Community Earth System Model version 2 (CESM2). We replaced the default parameterized $SO_2$ aqueous-phase reactions with detailed $HO_x$-, Fe-, N- and carbonate chemistry in cloud droplets and performed a global simulation for 2014-2015. Compared with the observations, the results incorporating detailed cloud aqueous-phase chemistry greatly reduced $SO_2$ overestimation. This overestimation was reduced by 0.1-10 ppbv in most of Europe, North America and Asia and more than 10 ppbv in parts of China. The biases in annual simulated $SO_2$ mixing ratios decreased by 46%, 41%, and 22% in Europe, the United States and China, respectively. Fe-chemistry and $HO_x$-chemistry contributed more to $SO_2$ oxidation than N-chemistry. Higher concentrations of soluble Fe and higher pH values could further enhance the oxidation capacity. This study emphasizes the importance of detailed in-cloud aqueous-phase chemistry for the oxidation of $SO_2$. These mechanisms can improve $SO_2$ simulation in CESM2 and deepen understanding of $SO_2$ oxidation and sulfate formation.

## 1  Introduction

Sulfur dioxide ($SO_2$) is one of the major atmospheric pollutants. The anthropogenic emission of $SO_2$ is the greatest source, which includes mainly the combustion of fossil fuel in the power and steel industries (Buchard et al., 2014). Human health risks from $SO_2$ have also been discovered and discussed in many studies (Kan et al., 2012; Tong et al., 2017; Chen et al., 2018). More importantly, $SO_2$ is the precursor of sulfate aerosols. Sulfate can be regarded as one of the core species in the atmosphere. Firstly, it is one of the major components of fine particles ($PM_{2.5}$), which cause haze pollution and affect human health, especially in East and South Asia (Buchard et al., 2014; Chen et al., 2018; Quan et al., 2015; Geng et al., 2019). In addition, sulfate is also the main component of cloud condensation nuclei (CCN), which directly influences the formation of clouds and thus affects precipitation, solar radiation and climate (He et al., 2015a; Tang et al., 2016). Moreover, sulfate itself is also one

of the key species affecting radiative forcing, which directly influences climate change (Li et al., 2018a; Pöschl and Shiraiwa, 2015; Xie et al., 2016). Therefore, only through a better understanding of $SO_2$, especially the process of its oxidation to sulfate, can we better understand sulfate and explore all the issues above (Hung et al., 2018).

$SO_2$ can be oxidized to sulfate in multiple ways. On clear and sunny days, the gas-phase oxidation of $SO_2$ by OH radicals ($\cdot$OH) is the dominant pathway (Li et al., 2018a; Cheng et al., 2016). However, when relative humidity (RH) and $PM_{2.5}$ increase on cloudy, foggy or hazy days, solar radiation and photochemical reactions decrease dramatically, resulting in a sharp decrease in gaseous $\cdot$OH and thus the gas-phase oxidation of $SO_2$, especially in winter. Alternatively, the aqueous-phase oxidation of $SO_2$ becomes much more important because of the increase in atmospheric liquid water content (Cheng et al., 2016; Quan et al., 2015). Aqueous-phase chemistry is an important part of atmospheric chemistry. Various physical and chemical parameters, such as the water content, ionic strength and pH value, could directly affect the gas-aqueous mass transfer process and the reaction rates and then influence the relative contributions of various mechanisms (Elser et al., 2016; Ervens, 2015). For $SO_2$, the aqueous-phase oxidation of $SO_2$ by diverse oxidants can serve as the major sink of atmospheric $SO_2$. It accounts for nearly 80% of global sulfate production, and more than half of sulfate production occurs in clouds (Harris et al., 2013; Huang et al., 2018). Specifically, there are several common oxidation pathways in the aqueous phase, such as oxidation by hydrogen peroxide ($H_2O_2$) and ozone ($O_3$) (Tan et al., 2016; Hung et al., 2018). In recent years, increasing numbers of studies have focused on the catalytic effect of transition metal ions (TMIs) on the aqueous-phase oxidation of $SO_2$ (Tilgner et al., 2013; Alexander et al., 2009). In addition, oxidation by $NO_2$ has also received increasing attention (Xue et al., 2016).

Transition metals in dust particles are important sites for various reactions and affect the moisture absorption, light scattering and nucleation process of clouds. Among these elements, Fe is one of the most important transition metals due to its high abundance and activity (Tang et al., 2016). Soluble Fe can act as an important catalyst in the Fenton reaction for the oxidation of $SO_2$ when dissolved into the aqueous phase. The Fenton reaction, which was firstly proposed by Henry J. H. Fenton in the 1890s, is one of the most important and widespread reactions in multiphase chemistry (Wiegand et al., 2017; Fenton, 1894; Pöschl and Shiraiwa, 2015). This reaction involves the production of $\cdot$OH in the aqueous phase by the decomposition of $H_2O_2$ catalyzed by low-valence TMIs such as $Fe^{2+}$ (Deguillaume et al., 2005; Herrmann et al., 2015). Different mechanisms have been developed to explain the first step of Fenton reactions. Two of the best known pathways are (1) the OH radical mechanism ($Fe^{2+} + H_2O_2 \rightarrow Fe^{3+} + \cdot OH + OH^-$) developed by Haber and Weiss and (2) the non-OH radical mechanism ($Fe^{2+} + H_2O_2 \rightarrow FeO^{2+} + H_2O$) proposed by Bray and Gorin (Fritz and Joseph, 1934; Bray and Gorin, 1932; Wiegand et al., 2017; Pöschl and Shiraiwa, 2015). The relative contributions of these two pathways differ under various conditions and remain controversial. A recent experimental study suggested that the non-OH radical mechanism is dominant under nearly neutral conditions (pH $\approx$ 7), while the OH radical mechanism becomes more important under acidic conditions (Pang et al., 2011; Deguillaume et al., 2005; Wiegand et al., 2017; Pöschl and Shiraiwa, 2015; Bataineh et al., 2012). Then, all of these oxidative intermediates (i.e., $Fe^{3+}$, $\cdot$OH and $FeO^{2+}$) can further oxidize $SO_2$ to sulfate. In this way, $Fe^{3+}$ and $FeO^{2+}$ are reduced to $Fe^{2+}$, thus forming a

complete redox cycle. Their concentrations and proportions are basically the same during the redox cycle, and a balance of catalysts is achieved (Deguillaume et al., 2005). The effects of soluble Fe on sulfate formation have been discussed in several studies (Gankanda et al., 2016). In addition, direct oxidation of $SO_2$ by $O_2$ might also be catalyzed by soluble Fe. In general, Fenton reactions could lead to faster radical recycling. The reaction rates of sulfate formation are enhanced with high Fe concentrations, especially when pH < 5 (Shao et al., 2019; Ervens, 2015; Huang et al., 2014; Tilgner et al., 2013).

On the other hand, a number of studies have also emphasized the important role of $NO_2$ in the oxidation of $SO_2$ (Ma et al., 2018; Tao et al., 2017; Huang et al., 2019). He et al. (2014) and Cheng et al. (2016) reported a missing source of $SO_2$ oxidation that can be explained by the synergistic effect between $NO_2$ and $SO_2$ in aerosol water and on mineral dust: $2\ NO_2 + HSO_3^- + H_2O \rightarrow 3\ H^+ + 2\ NO_2^- + SO_4^{2-}$ (He et al., 2014; Cheng et al., 2016). Such a conversion of $SO_2$ by $NO_2$ is driven by a high pH value (e.g., pH > 5.5) and a high concentration of $NO_2$ (Wang et al., 2020; Li et al., 2018b; Huang et al., 2019; He and He, 2020; He et al., 2018). Moreover, these studies have indicated that > 95% of $NO_2$ converts to $HNO_2/NO_2^-$ by hitting the surface of $NaHSO_3$ aqueous microjets to promote the aqueous-phase oxidation of $SO_2$ (Wang et al., 2020; Li et al., 2018b). This pathway can explain the gaps in sulfate concentrations between simulations and observations from approximately 15% to 65% during haze days in winter (Zheng et al., 2020). However, other studies have suggested that the contribution of nitrogen chemistry to $SO_2$ oxidation is very limited. Au Yang et al. (2018) argued that the $NO_2$ oxidation pathway cannot explain the extreme concentrations of sulfate measured in urban aerosols (Au Yang et al., 2018). Only a minor (approximately 2%) fraction of heterogeneous sulfate formation occurs via oxidation of $SO_2$ by $NO_2$ (Shao et al., 2019). The main reason is that the pH value is hardly ever high enough to maintain the efficiency of oxidation by $NO_2$ in aerosol or cloud water (Guo et al., 2017). For instance, aerosols collected from several urban areas in China (CN) were always acidic (even with the unusually high $NH_3$ emissions and concentrations in northern CN), suggesting that oxidation by $NO_2$ might not be very important in these regions (Li et al., 2020; He and He, 2020). In summary, the contribution of N-chemistry to the aqueous-phase oxidation of $SO_2$ still needs further investigation.

Many studies have been conducted on the aqueous-phase oxidation of $SO_2$. Some laboratory studies have focused on the detailed mechanism, such as the radical processes involved in different pathways of the Fenton reaction (Wiegand et al., 2017; Bataineh et al., 2012) and the conversion of $NO_2$ to $HNO_2$ to oxidize $SO_2$ (He et al., 2014). Some studies have paid more attention to the measurement and updating of kinetic parameters (Cwiertny et al., 2008; He et al., 2018; He and He, 2020). More importantly, modelling studies have made great progress in revealing the mechanism of $SO_2$ oxidation and sulfate formation in the aqueous phase (Bell et al., 2005). For instance, Herrmann et al. (2000) used a box model to investigate the detailed aqueous-phase radical mechanism for tropospheric chemistry (Herrmann et al., 2000). Huang et al. (2014) and Li et al. (2017) discussed the enhancement of sulfate formation by mineral aerosols in CN and improved the simulation of heterogeneous sulfate in the WRF-Chem model (Huang et al., 2014; Li et al., 2017). Li et al. (2018a) also improved the simulation of sulfate with the NAQPMS model of oxidation of $SO_2$ by $NO_2$ on wet aerosols on haze days (Li et al., 2018a).

Shao et al. (2019) evaluated various heterogeneous mechanisms for sulfate aerosol formation in Beijing using the GEOS-Chem model (Shao et al., 2019). Bell et al. (2005) analyzed the effects of different $SO_2$ emission scenarios on radiative forcing and climate over East Asia (EA) using CESM2 (Bell et al., 2005). Both Zheng et al. (2020) and Zheng et al. (2015) used the CMAQ model to explore heterogeneous chemistry for the formation of secondary inorganic aerosols and the contribution of nitrate photolysis to heterogeneous sulfate formation in CN on winter haze days, respectively (Zheng et al., 2020; Zheng et al., 2015).

Zheng et al. (2015) used the WRF-CMAQ model to explain the crucial role of reactive N-chemistry in aerosol water for sulfate formation during haze events in CN (Zheng et al., 2015). Nevertheless, there are still obvious shortcomings in these model studies. First of all, in long-term global climate simulations, studies focused on the spatio-temporal distribution of $SO_2$ concentrations are still insufficient. Most studies have evaluated only sulfate distribution and its climate impact. Very few studies have discussed the simulation of $SO_2$ and these few only from the perspective of $SO_2$ emissions. In addition, although

some studies have attempted to discuss different pathways of aqueous-phase oxidation of $SO_2$, most of them have merely adopted simplified mechanisms or even parameterization alone without introducing detailed radical mechanisms. On the other hand, several studies investigated the detailed aqueous-phase chemistry, but they did not analyze its influence on $SO_2$ but discussed that on only $O_3$, $\cdot OH$ or $HO_2$ (Herrmann et al., 2000; Jacob, 1986; Matthijsen et al., 1995; Jacob, 2000; Mao et al., 2013; Mao et al., 2017). Finally, the simulations of $SO_2$ in many studies are still highly overestimated (He et al., 2015b; He et

al., 2015a; Buchard et al., 2014; Hong et al., 2017; Georgiou et al., 2018; Wei et al., 2019; Flemming et al., 2015; Sha et al., 2019; Liu et al., 2012b; Hedegaard et al., 2008), while others underestimate the concentration of sulfate (Xie et al., 2016; Goto et al., 2015; Bell et al., 2005; Lamarque et al., 2012; Pozzer et al., 2012; Guth et al., 2016; Wei et al., 2019; Geng et al., 2019; Kajino et al., 2012; Mathur, 2005; Liu et al., 2012b; Sha et al., 2019; Zhang et al., 2012; Itahashi, 2018). All of these disadvantages indicate that the mechanism of $SO_2$ oxidation to sulfate is still not fully understood.

This study aims to examine the role played by detailed in-cloud aqueous-phase chemistry (not including chemical reactions on the surfaces of wet aerosols) on the capacity for oxidation of global $SO_2$ in the Community Earth System Model 2 (CESM2). We describe the CESM2 model, detailed cloud chemistry and observational data in Sect. 2. The evaluation of $SO_2$ simulations with or without coupling detailed in-cloud aqueous-phase chemistry is given in Sect. 3. The contributions of different in-cloud aqueous-phase chemical mechanisms to the simulation of $SO_2$ are analyzed in Sect. 4. The key factors that affect the capacity

for $SO_2$ oxidation are discussed in Sect. 5. Finally, the main conclusions are drawn in Sect. 6.

## 2   Methodology

### 2.1   Model description

The Community Earth System Model 2 (CESM2, v2.1.1), developed by the National Center for Atmospheric Research (NCAR, https://www.cesm.ucar.edu/models/cesm2/, last access: 16 December 2020) is used in this study (Emmons et al., 2020;

Danabasoglu et al., 2020), configured with the Community Atmosphere Model version 4.0 (CAM4). The coupled chemistry in CAM4 is primarily based on the Model for Ozone and Related chemical Tracers, version 4 (MOZART-4), including 85 gas-phase species with bulk aerosols and detailed tropospheric chemistry with 196 gas-phase reactions (Emmons et al., 2010; Lamarque et al., 2012). The default aerosol species simulated in this component set include sulfate, nitrate, ammonium, black carbon (BC), organic carbon (OC), secondary organic aerosol (SOA), dust and sea salt. In this study, we develop a detailed aqueous-phase chemistry module for $SO_2$ oxidation fully coupled in the MOZART-4 chemistry.

The model is configured with a horizontal resolution of 0.95° (latitude) × 1.25° (longitude) and 30 levels in the vertical direction from 993 (near-surface layer) to 3.6 hPa. The model is nudged by assimilated meteorological offline data from Modern-Era Retrospective analysis for Research and Applications, version 2 (MERRA2, https://rda.ucar.edu/datasets/ds313.3/, last access: 20 July 2020), prepared with 14 meteorological variables (e.g., air temperature, surface pressure, specific humidity and eastward and northward winds) to run CESM2 simulations. The meteorological data have a temporal resolution of 3 h.

All the emission inventories needed for MOZART-4 chemistry are obtained from the CESM database (https://svn-ccsm-inputdata.cgd.ucar.edu/trunk/inputdata/atm/cam/chem/CMIP6_emissions_1750_2015, last access: 31 December 2020 ), which was developed for the CMIP6 projects (Feng et al., 2020). The inventories have been updated to 2015, which is the year of the simulation in this study. Meanwhile, the emission, dry deposition and wet deposition processes of aerosol species are also guided by input files from CESM database (https://svn-ccsm-inputdata.cgd.ucar.edu/trunk/inputdata/atm/cam/chem/trop_mozart_aero/ ; https://svn-ccsm-inputdata.cgd.ucar.edu/trunk/inputdata/atm/cam/chem/emis/CMIP6_emissions_1750_2015_2deg/) and the source codes of CESM2 (aero_model.F90, mo_drydep.F90 and wetdep.F90).

The variables related to the cloud properties used in this study are all from the Rasch and Kristjansson (RK) prognostic cloud microphysical processes. These variables include the liquid water content of clouds (LWC, $L_{water}$ $L_{air}^{-1}$), volume fraction of clouds ($F_{cld}$) and radius of cloud droplets (r, μm). They are directly obtained from the model simulation and directly or indirectly influence the in-cloud aqueous-phase chemistry. Among these variables, the simulated r ranges from 8 μm to 14 μm, consistent with those in previous studies (Herrmann et al., 2000; Jacob, 1986; Matthijsen et al., 1995; Liu et al., 2012a; Herrmann et al., 2015). Meanwhile, CESM2 simulates both large-scale stratiform clouds and convective clouds (i.e., shallow cumulus clouds and deep convective clouds). For each type of cloud, both water and ice are simulated. However, the $SO_2$ produced in convective clouds is assumed to be removed rapidly by convective precipitation. Thus, the contribution of $SO_2$ from shallow cumulus clouds and deep convective clouds is ignored. Only the LWC and $F_{cld}$ of large-scale liquid stratiform clouds are employed in this study.

## 2.2 Mechanism of in-cloud aqueous-phase oxidation of $SO_2$

The detailed mechanism of in-cloud aqueous-phase oxidation of $SO_2$ is divided into the gas-aqueous phase transfer process and aqueous-phase chemical mechanisms, listed in Tables 1a and 1b, respectively (see below). There are 32 (16 pairs of) gas-aqueous phase transfer equilibria and 187 aqueous-phase reactions (only in cloud droplets, not on surfaces of wet aerosol), involving 46 new aqueous species in all. Specifically, the aqueous-phase reactions include 26 (13 pairs of) ionization equilibria and four different chemistry modules, which are $HO_x$-chemistry, Fe-chemistry, N-chemistry and carbonate chemistry. The

aqueous-phase oxidation of $SO_2$ by $H_2O_2$ and $O_3$ is included in the $HO_x$-chemistry mechanism. The two pathways of the Fenton reaction are included in the Fe-chemistry mechanism. The aqueous-phase oxidation of $SO_2$ by $NO_2$ is included in the N-chemistry mechanism.

**Table 1a: Gas-aqueous phase transfer equilibria.**

| No. | Reactions | $k_1$ | $k_2$ | Reference |
|---|---|---|---|---|
| | Gas-aqueous phase transfer | | | |
| 1[a, c] | $O_3(g) \rightarrow O_3$ | 48 | 0.05 | (Mirabel, 1996) |
| 2[b] | $O_3 \rightarrow O_3(g)$ | $1.1 \times 10^{-2}$ | -2397 | (Hoffman and Calvert, 1985; Pandis and Seinfeld, 1989) |
| 3[a] | $HO_2(g) \rightarrow HO_2$ | 33 | 0.01 | (Hanson et al., 1992) |
| 4[b] | $HO_2 \rightarrow HO_2(g)$ | $9.0 \times 10^3$ | 0 | (Weinsteinlloyd and Schwartz, 1991) |
| 5[a] | $OH(g) \rightarrow OH$ | 17 | 0.05 | (Herrmann et al., 2000) |
| 6[b] | $OH \rightarrow OH(g)$ | 25 | -5280 | (Kläning et al., 1985) |
| 7[a] | $H_2O_2(g) \rightarrow H_2O_2$ | 34 | 0.23 | (Seinfeld and Pandis, 2016) |
| 8[b] | $H_2O_2 \rightarrow H_2O_2(g)$ | $1.02 \times 10^5$ | -6339 | (Lind and Kok, 1994) |
| 9[a] | $SO_2(g) \rightarrow SO_2$ | 64 | $[1+\exp(14.7-3825/T)]^{-1}$ | (Boniface et al., 2000) |
| 10[b] | $SO_2 \rightarrow SO_2(g)$ | 1.2 | -3157 | (Olson and Hoffmann, 1989) |
| 11[a] | $CO_2(g) \rightarrow CO_2$ | 44 | $2 \times 10^{-4}$ | (Herrmann et al., 2000) |
| 12[b] | $CO_2 \rightarrow CO_2(g)$ | $3.11 \times 10^{-2}$ | -2422 | (Chameides, 1984) |
| 13[a] | $NH_3(g) \rightarrow NH_3$ | 17 | 0.04 | (Bongartz et al., 1995) |
| 14[b] | $NH_3 \rightarrow NH_3(g)$ | 60.7 | -3921 | (Clegg and Brimblecombe, 1990) |
| 15[a] | $HNO_3(g) \rightarrow HNO_3$ | 63 | 0.054 | (Davidovits et al., 1995) |
| 16[b] | $HNO_3 \rightarrow HNO_3(g)$ | $2.1 \times 10^5$ | -8696 | (Lelieveld and Crutzen, 1991) |
| 17[a] | $HCOOH(g) \rightarrow HCOOH$ | 46 | 0.012 | (Davidovits et al., 1995) |
| 18[b] | $HCOOH \rightarrow HCOOH(g)$ | $5.53 \times 10^3$ | -5629 | (Khan and Brimblecombe, 1992) |
| 19[a] | $CH_3COOH(g) \rightarrow CH_3COOH$ | 60 | 0.019 | (Davidovits et al., 1995) |
| 20[b] | $CH_3COOH \rightarrow CH_3COOH(g)$ | $5.50 \times 10^3$ | -5894 | (Khan and Brimblecombe, 1992) |
| 21[a] | $NO_3(g) \rightarrow NO_3$ | 62 | $4 \times 10^{-3}$ | (Kirchner et al., 1990; Rudich et al., 1996) |
| 22[b] | $NO_3 \rightarrow NO_3(g)$ | 0.6 | 0 | (Rudich et al., 1996) |
| 23[a] | $N_2O_5(g) \rightarrow N_2O_5$ | 108 | $3.7 \times 10^{-3}$ | (George et al., 1994) |
| 24[b] | $N_2O_5 \rightarrow N_2O_5(g)$ | 1.4 | 0 | (Herrmann et al., 2000) |
| 25[a] | $NO_2(g) \rightarrow NO_2$ | 46 | $2 \times 10^{-4}$ | (Shao et al., 2019) |
| 26[b] | $NO_2 \rightarrow NO_2(g)$ | $1.0 \times 10^{-2}$ | -2518 | (Sander, 1999; Pandis and Seinfeld, 1989) |

| 27[a] | $HO_2NO_2(g) \rightarrow HO_2NO_2$ | 79 | 0.1 | (Jacob, 1986) |
|---|---|---|---|---|
| 28[b] | $HO_2NO_2 \rightarrow HO_2NO_2(g)$ | $1 \times 10^5$ | 0 | (Herrmann et al., 2000) |
| 29[a] | $NO(g) \rightarrow NO$ | 30 | 0.1 | Estimated |
| 30[b] | $NO \rightarrow NO(g)$ | $1.9 \times 10^{-3}$ | -1460 | (Sander, 1999; Pandis and Seinfeld, 1989) |
| 31[a] | $O_2(g) \rightarrow O_2$ | 32 | 0.1 | Estimated |
| 32[b] | $O_2 \rightarrow O_2(g)$ | $1.3 \times 10^{-3}$ | 0 | (Sander, 1999) |

[a] Reaction rate constant $k = \frac{3 D_g LWC}{\Lambda r^2}$. The unit is s⁻¹. Gas phase diffusion coefficient $D_g = \frac{9.45 \times 10^{17}}{[M]} \sqrt{T(0.03472 + \frac{1}{k_1})}$. LWC is the volume

mixing ratio of cloud liquid water. $\Lambda = 1 + \left(\lambda + 1.3 \left(\frac{1}{k_2} - 1\right)\right)$, $\lambda = \frac{0.71 + 1.3\beta}{1 + \beta}$, $\beta = 4.54 \times 10^{-15} \sqrt{V_g^2 + V_{air}^2}$, $V_g = \sqrt{\frac{8RT}{\pi k_1}}$, $V_{air} =$

$\sqrt{\frac{8RT}{28.8\pi}}$, $R = 8.31 \times 10^7$ is the ideal gas constant (multiplied by a factor to keep $V_g$ and $V_{air}$ in the unit of cm s⁻¹), r is the radius of cloud

droplets in cm, [M] is the number density of air in the unit of molecules cm⁻³. T is atmospheric temperature in Kelvin. $k_1$ is the molar mass

(g mol⁻¹). $k_2$ is the mass accommodation coefficients. All the formulas above refer to (Shao et al., 2019; Liang and Jacobson, 1999).

[b] Reaction rate constant $k = \frac{k_{n-1}}{0.082\ T\ LWC\ C}$. The unit is s⁻¹. $C = k_1 exp\left(-k_2 \left(\frac{1}{T} - \frac{1}{298}\right)\right)$, $k_{n-1}$ is the rate constant of its reverse reaction

with [a]. LWC is as in [a]. $k_1$ is Henry's law constants (M atm⁻¹) at 298 K. $k_2$ is ΔH (J mol⁻¹) / R (J mol⁻¹ K⁻¹). ΔH is the enthalpy of dissolution.
All the formulas above refer to (Liang and Jacobson, 1999).

[c] All species are liquid species by default, and gas species are marked with (g). The same below.

**Table 1b. Aqueous-phase chemistry.**

| No. | Reactions | $k_{298}$, M⁻ⁿ s⁻¹ [a] | $E_a/R$, K | Reference |
|---|---|---|---|---|
| | Aqueous ionization equilibria | | | |
| 33 | $H_2O_2 \rightarrow H^+ + HO_2^-$ | $1.26 \times 10^{-2}$ | | (De Laat and Le, 2005) |
| 34 | $H^+ + HO_2^- \rightarrow H_2O_2$ | $10^{10}$ | | (De Laat and Le, 2005) |
| 35 | $HO_2 \rightarrow H^+ + O_2^-$ | $1.14 \times 10^6$ | | (Miller et al., 2013) |
| 36 | $H^+ + O_2^- \rightarrow HO_2$ | $7.2 \times 10^{10}$ | | (Miller et al., 2013) |
| 37 | $CO_2 + H_2O \rightarrow H^+ + HCO_3^-$ | $3.84 \times 10^4$ | 9250 | (Welch et al., 1969; Graedel and Weschler, 1981) |
| 38 | $H^+ + HCO_3^- \rightarrow CO_2 + H_2O$ | $5 \times 10^{10}$ | | (Graedel and Weschler, 1981) |
| 39 | $HCO_3^- \rightarrow H^+ + CO_3^{2-}$ | 2.35 | 1820 | (Harned and Owen, 1958) |
| 40 | $H^+ + CO_3^{2-} \rightarrow HCO_3^-$ | $5 \times 10^{10}$ | | (Graedel and Weschler, 1981) |
| 41 | $NH_3 + H_2O \rightarrow NH_4^+ + OH^-$ | $6.02 \times 10^5$ | 560 | (Harned and Owen, 1958) |
| 42 | $NH_4^+ + OH^- \rightarrow NH_3 + H_2O$ | $3.4 \times 10^{10}$ | | (Graedel and Weschler, 1981) |
| 43 | $HNO_3 \rightarrow H^+ + NO_3^-$ | $1.1 \times 10^{12}$ | -1800 | (Redlich, 1946) |
| 44 | $H^+ + NO_3^- \rightarrow HNO_3$ | $5 \times 10^{10}$ | | (Graedel and Weschler, 1981) |
| 45 | $HNO_2 \rightarrow H^+ + NO_2^-$ | $2.65 \times 10^7$ | 1760 | (Park and Lee, 1988) |
| 46 | $H^+ + NO_2^- \rightarrow HNO_2$ | $5 \times 10^{10}$ | | (Graedel and Weschler, 1981) |
| 47 | $HO_2NO_2 \rightarrow H^+ + O_2NO_2^-$ | $5 \times 10^5$ | | (Lammel et al., 1990) |
| 48 | $H^+ + O_2NO_2^- \rightarrow HO_2NO_2$ | $5 \times 10^{10}$ | | (Herrmann et al., 2000) |
| 49 | $SO_2 + H_2O \rightarrow H^+ + HSO_3^-$ | $6.27 \times 10^4$ | -1940 | (Beilke and Gravenhorst, 1978; Harned and Owen, 1958) |
| 50 | $H^+ + HSO_3^- \rightarrow SO_2 + H_2O$ | $2.0 \times 10^8$ | | (Graedel and Weschler, 1981) |
| 51 | $HSO_3^- \rightarrow H^+ + SO_3^{2-}$ | 3110 | -1960 | (Beilke and Gravenhorst, 1978) |
| 52 | $H^+ + SO_3^{2-} \rightarrow HSO_3^-$ | $5 \times 10^{10}$ | | (Graedel and Weschler, 1981) |
| 53 | $HSO_4^- \rightarrow H^+ + SO_4^{2-}$ | $1.02 \times 10^9$ | -2700 | (Redlich, 1946) |

| 54 | $H^+ + SO_4^{2-} \rightarrow HSO_4^-$ | $1 \times 10^{11}$ | | (Graedel and Weschler, 1981) |
| 55 | $HCOOH \rightarrow H^+ + HCOO^-$ | $8.85 \times 10^6$ | -12 | (Harned and Owen, 1958) |
| 56 | $H^+ + HCOO^- \rightarrow HCOOH$ | $5 \times 10^{10}$ | | (Graedel and Weschler, 1981) |
| 57 | $CH_3COOH \rightarrow H^+ + CH3COO^-$ | $8.75 \times 10^5$ | -46 | (Harned and Owen, 1958) |
| 58 | $H^+ + CH3COO^- \rightarrow CH3COOH$ | $5 \times 10^{10}$ | | (Graedel and Weschler, 1981) |

**HO$_x$-chemistry**

| 59 | $H_2O_2 \xrightarrow{h\nu} 2\ OH$ | See ref. | | (Zellner et al., 1990) |
| 60 | $O_3 \xrightarrow{H_2O,h\nu} H_2O_2 + O_2$ | See ref. | | (Graedel and Weschler, 1981) |
| 61 | $OH + HO_2 \rightarrow H_2O + O_2$ | $6.6 \times 10^9$ | 1500 | (Sehested et al., 1968; Thomas, 1963) |
| 62 | $HO_2 + HO_2 \rightarrow H_2O_2 + O_2$ | $8.3 \times 10^5$ | 2700 | (Bielski et al., 1985) |
| 63 | $OH + H_2O_2 \rightarrow HO_2 + H_2O$ | $2.7 \times 10^7$ | 1700 | (Christensen et al., 1982; Buxton et al., 1988b) |
| 64 | $O_2^- + O_3 \xrightarrow{H_2O} OH + OH^- + 2\ O_2$ | $1.5 \times 10^9$ | 1500 | (Sehested et al., 1983; Bielski et al., 1985) |
| 65 | $OH + HSO_3^- \xrightarrow{O_2} SO_5^- + H_2O$ | $4.5 \times 10^9$ | 1500 | (Huie and Neta, 1987) |
| 66 | $OH + SO_3^{2-} \xrightarrow{O_2} SO_5^- + OH^-$ | $5.5 \times 10^9$ | 1500 | (Huie and Neta, 1987; Adams and Boag, 1964; Buxton et al., 1988b) |
| 67 | $HCOO^- + OH \xrightarrow{O_2} CO_2 + HO_2 + OH^-$ | $3.2 \times 10^9$ | 1250 | (Chin and Wine, 1994) |
| 68 | $SO_3^{2-} + SO_4^- \xrightarrow{O_2} SO_4^{2-} + SO_5^-$ | $7.5 \times 10^8$ | 1500 | (Wine et al., 1989) |
| 69 | $HSO_3^- + SO_4^- \xrightarrow{O_2} SO_4^{2-} + SO_5^- + H^+$ | $7.5 \times 10^8$ | 1500 | (Wine et al., 1989) |
| 70 | $HSO_3^- + O_3 \rightarrow SO_4^{2-} + H^+ + O_2$ | $3.7 \times 10^5$ | 5530 | (Hoffmann, 1986; Wine et al., 1989) |
| 71 | $SO_3^{2-} + O_3 \rightarrow SO_4^{2-} + O_2$ | $1.5 \times 10^9$ | 5280 | (Hoffmann, 1986; Wine et al., 1989) |
| 72 | $SO_4^- + OH^- \rightarrow SO_4^{2-} + OH$ | $8.0 \times 10^7$ | 1500 | (Maruthamuthu and Neta, 1978) |
| 73 | $SO_4^- + H_2O_2 \rightarrow H^+ + SO_4^{2-} + HO_2$ | $1.2 \times 10^7$ | 2000 | (Wine et al., 1989) |
| 74 | $SO_4^- (+ H_2O) \rightarrow SO_4^{2-} + H^+ + OH$ | 440 | 1850 | (Bao and Barker, 1996) |
| 75 | $SO_4^- + HCOO^- \xrightarrow{O_2} SO_4^{2-} + CO_2 + HO_2$ | $1.1 \times 10^8$ | 1500 | (Reese et al., 1997; Wine et al., 1989) |
| 76 | $HCOOH + OH \xrightarrow{O_2} H_2O + CO_2 + HO_2$ | $1.1 \times 10^8$ | 1000 | (Chin and Wine, 1994) |
| 77 | $O_3 + H_2O_2 + OH^- \rightarrow OH + O_2^- + O_2 + H_2O$ | $4.4 \times 10^8$ | -4000 | (Staehelin and Hoigne, 1982) |
| 78 | $SO_4^- + HO_2 \rightarrow SO_4^{2-} + H^+ + O_2$ | $5.0 \times 10^9$ | 1500 | (Jacob, 1986) |
| 79 | $SO_4^- + O_2^- \rightarrow SO_4^{2-} + O_2$ | $5.0 \times 10^9$ | 1500 | (Jacob, 1986) |
| 80 | $HCOO^- + O_3 \rightarrow CO_2 + OH + O_2^-$ | $1.0 \times 10^2$ | 5500 | (Hoigne and Bader, 1983b) |
| 81 | $SO_5^- + HCOO^- \xrightarrow{O_2} HSO_5^- + CO_2 + O_2^-$ | $1.4 \times 10^4$ | 4000 | (Jacob, 1986) |
| 82 | $SO_5^- + HSO_3^- \xrightarrow{O_2} HSO_5^- + SO_5^-$ | $2.5 \times 10^4$ | 3850 | (Huie and Neta, 1987) |
| 83 | $HSO_5^- + OH \rightarrow SO_5^- + H_2O$ | $1.7 \times 10^7$ | 1900 | (Maruthamuthu and Neta, 1977) |
| 84 | $HSO_5^- + HSO_3^- + H^+ \rightarrow 2\ SO_4^{2-} + 3\ H^+$ | $1.7 \times 10^7$ | 2000 | (Mcelroy, 1987; Betterton and Hoffmann, 1988) |
| 85 | $SO_5^- + HSO_3^- \rightarrow SO_4^- + SO_4^{2-} + H^+$ | $7.5 \times 10^4$ | 3500 | (Huie and Neta, 1987) |
| 86 | $O_2^- + SO_5^- \xrightarrow{H_2O} O_2 + HSO_5^- + OH^-$ | $1.0 \times 10^8$ | 1050 | (Jacob, 1986) |
| 87 | $OH + HSO_3^- \xrightarrow{O_2} SO_4^{2-} + H^+ + HO_2$ | $4.5 \times 10^9$ | | (Huie and Neta, 1987) |
| 88 | $OH + O_3 \rightarrow HO_2 + O_2$ | $2.0 \times 10^9$ | | (Buhler et al., 1984) |
| 89 | $HO_2 + O_2^- \xrightarrow{H^+} H_2O_2 + O_2$ | $9.7 \times 10^7$ | 1060 | (Bielski et al., 1985) |
| 90 | $O_2^- + OH \rightarrow OH^- + O_2$ | $1.1 \times 10^{10}$ | 2120 | (Christensen et al., 1989) |

| | | | | |
|---|---|---|---|---|
| 91 | $HSO_3^- + OH \rightarrow H_2O + SO_3^-$ | $2.7 \times 10^9$ | | (Buxton et al., 1996b) |
| 92 | $SO_3^{2-} + OH \rightarrow OH^- + SO_3^-$ | $4.6 \times 10^9$ | | (Buxton et al., 1996b) |
| 93 | $HSO_3^- + H_2O_2 + H^+ \rightarrow SO_4^{2-} + H_2O + 2\ H^+$ | $6.9 \times 10^7$ | 4000 | (Lind et al., 1987) |
| 94 | $SO_2 + O_3 \xrightarrow{H_2O} HSO_4^- + O_2 + H^+$ | $2.4 \times 10^4$ | | (Hoffmann, 1986) |
| 95 | $SO_5^- + SO_5^- \rightarrow S_2O_8^{2-} + O_2$ | $1.8 \times 10^8$ | 2600 | (Herrmann et al., 1995) |
| 96 | $SO_5^- + SO_5^- \rightarrow 2\ SO_4^- + O_2$ | $7.2 \times 10^6$ | 2600 | (Herrmann et al., 1995) |
| 97 | $SO_5^- + HO_2 \rightarrow HSO_5^- + O_2$ | $1.7 \times 10^9$ | | (Buxton et al., 1996a) |
| 98 | $SO_3^- + O_2 \rightarrow SO_5^-$ | $2.5 \times 10^9$ | | (Buxton et al., 1996b) |
| 99 | $SO_5^- + HSO_3^- \rightarrow HSO_5^- + SO_3^-$ | $8.6 \times 10^3$ | | (Buxton et al., 1996b) |
| 100 | $SO_5^- + SO_3^{2-} \xrightarrow{H^+} HSO_5^- + SO_3^-$ | $2.13 \times 10^5$ | | (Buxton et al., 1996b) |
| 101 | $SO_5^- + SO_3^{2-} \rightarrow SO_4^- + SO_4^{2-}$ | $5.5 \times 10^5$ | | (Buxton et al., 1996b) |
| 102 | $OH + HSO_4^- \rightarrow H_2O + SO_4^-$ | $3.5 \times 10^5$ | | (Tang et al., 1988) |
| 103 | $SO_4^- + HSO_3^- \rightarrow SO_4^{2-} + SO_3^- + H^+$ | $3.2 \times 10^8$ | | (Reese et al., 1997) |
| 104 | $SO_4^- + SO_3^{2-} \rightarrow SO_4^{2-} + SO_3^-$ | $3.2 \times 10^8$ | 1200 | (Reese et al., 1997) |
| 105 | $HSO_5^- + SO_3^{2-} + H^+ \rightarrow 2\ SO_4^{2-} + 2\ H^+$ | $7.14 \times 10^6$ | | (Betterton and Hoffmann, 1988) |
| 106 | $HCOOH + SO_4^- \xrightarrow{O_2} SO_4^{2-} + H^+ + HO_2 + CO_2$ | $2.5 \times 10^6$ | | (Reese et al., 1997) |
| 107 | $O_2^- + H_2O_2 \rightarrow OH^- + OH + O_2$ | 0.13 | | (Bielski et al., 1985) |
| 108 | $OH + OH \rightarrow H_2O_2$ | $5.5 \times 10^9$ | | (Miller et al., 2013) |
| 109 | $H_2O_2 + HO_2 \rightarrow H_2O + O_2 + OH$ | 3.1 | | (Miller et al., 2013) |
| 110 | $HO_2 + O_2^- \rightarrow HO_2^- + O_2$ | $9.7 \times 10^7$ | | (De Laat and Le, 2005) |
| 111 | $O_2^{2-} + H^+ \rightarrow HO_2^-$ | $10^{10}$ | | (De Laat and Le, 2005) |
| 112 | $SO_4^- + SO_4^- \rightarrow S_2O_8^{2-}$ | $4.5 \times 10^8$ | | (Buxton et al., 1996b) |
| 113 | $OH^- + O_3 \xrightarrow{H_2O} H_2O_2 + O_2 + OH^-$ | 70 | | (Staehelin and Hoigne, 1982) |
| 114 | $HO_2^- + O_3 \rightarrow OH + O_2^- + O_2$ | $2.8 \times 10^6$ | 2500 | (Staehelin and Hoigne, 1982) |
| 115 | $H_2O_2 + O_3 \rightarrow H_2O + 2\ O_2$ | $7.8 \times 10^{-3}\ [O_3]^{-0.5}$ | | (Martin et al., 1981) |
| 116 | $HCOOH + O_3 \rightarrow CO_2 + HO_2 + OH$ | 5.0 | 0 | (Hoigne and Bader, 1983a) |
| 117 | $SO_2 + H_2O_2 \xrightarrow{H_2O} SO_4^{2-} + 2\ H^+ + H_2O$ | $7.5 \times 10^7$ | 4430 | (Mcardle and Hoffmann, 1983) |
| 118 | $SO_3^{2-} + H_2O_2 \rightarrow SO_4^{2-} + H_2O$ | $7.5 \times 10^7$ | 4430 | (Mcardle and Hoffmann, 1983) |
| 119 | $SO_5^- + SO_3^{2-} \xrightarrow{O_2,H_2O} HSO_5^- + SO_5^- + OH^-$ | $2.5 \times 10^4$ | 2000 | (Huie and Neta, 1987) |
| 120 | $SO_5^- + HCOOH \xrightarrow{O_2} HSO_5^- + CO_2 + HO_2$ | 200 | 5300 | (Jacob, 1986) |
| 121 | $SO_2 + HO_2 \xrightarrow{H_2O} SO_4^{2-} + OH + 2\ H^+$ | $1.0 \times 10^6$ | 0 | (Hoffman and Calvert, 1985) |
| 122 | $HSO_3^- + HO_2 \rightarrow SO_4^{2-} + OH + H^+$ | $1.0 \times 10^6$ | 0 | (Hoffman and Calvert, 1985) |
| 123 | $SO_3^{2-} + HO_2 \rightarrow SO_4^{2-} + OH$ | $1.0 \times 10^6$ | 0 | (Hoffman and Calvert, 1985) |
| 124 | $SO_2 + O_2^- \xrightarrow{H_2O} SO_4^{2-} + OH + H^+$ | $1.0 \times 10^5$ | 0 | (Hoffman and Calvert, 1985) |
| 125 | $HSO_3^- + O_2^- \rightarrow SO_4^{2-} + OH$ | $1.0 \times 10^5$ | 0 | (Hoffman and Calvert, 1985) |
| 126 | $SO_3^{2-} + O_2^- \xrightarrow{H_2O} SO_4^{2-} + OH + OH^-$ | $1.0 \times 10^5$ | 0 | (Hoffman and Calvert, 1985) |
| 127 | $SO_3^- + SO_3^- \xrightarrow{H_2O} SO_3^{2-} + H^+ + HSO_4^-$ | 0.37 | | (Fischer and Warneck, 1996) |

**Fe-chemistry**

| | | | | |
|---|---|---|---|---|
| 128 | $FeOH^{2+} \xrightarrow{h\nu} Fe^{2+} + OH$ | See ref. | | (Benkelberg and Warneck, 1995) |
| 129 | $FeSO_4^+ \xrightarrow{h\nu} Fe^{2+} + SO_4^-$ | See ref. | | (Benkelberg and Warneck, 1995) |
| 130 | $H_2O_2 + Fe^{2+} \rightarrow FeOH^{2+} + OH$ | $63 + (3 \times 10^{-10}\ [H^+]^{-1}\ 5.9 \times 10^6)$ | | (Millero and Sotolongo, 1989) |
| 131 | $Fe^{2+} + O_3 \xrightarrow{H_2O} FeOH^{2+} + OH + O_2$ | $8.2 \times 10^5$ | | (Logager et al., 1992) |
| 132 | $FeOH^{2+} + HSO_3^- \xrightarrow{O_2} Fe^{2+} + SO_4^{2-} + H_2O$ | $[FeOH^{2+}] \times 1 \times 10^9$ | | (Martin et al., 1991) |
| 133 | $O_3 + Fe^{2+} \rightarrow FeO^{2+} + O_2$ | $8.2 \times 10^5$ | | (Logager et al., 1992) |
| 134 | $H_2O_2 + FeO^{2+} \rightarrow FeOH^{2+} + HO_2$ | $9.5 \times 10^3$ | 2800 | (Jacobsen et al., 1997) |

| No. | Reaction | Rate | E/R | Reference |
|---|---|---|---|---|
| 135 | $HO_2 + FeO^{2+} \rightarrow FeOH^{2+} + O_2$ | $2.0 \times 10^6$ | | (Jacobsen et al., 1997) |
| 136 | $OH + FeO^{2+} \xrightarrow{H_2O} FeOH^{2+} + H_2O_2$ | $1.0 \times 10^7$ | | (Logager et al., 1992; Jacobsen et al., 1997) |
| 137 | $FeO^{2+} + H_2O \rightarrow FeOH^{2+} + OH$ | $1.3 \times 10^{-2}$ | 4100 | (Jacobsen et al., 1997) |
| 138 | $FeO^{2+} + Fe^{2+} \xrightarrow{H_2O} 2\ FeOH^{2+}$ | $7.2 \times 10^4$ | 840 | (Jacobsen et al., 1997) |
| 139 | $FeO^{2+} + Fe^{2+} \xrightarrow{H_2O} Fe(OH)_2Fe^{4+}$ | $1.8 \times 10^4$ | 5050 | (Jacobsen et al., 1997) |
| 140 | $Fe(OH)_2Fe^{4+} \rightarrow 2\ FeOH^{2+}$ | 0.49 | 8800 | (Jacobsen et al., 1997) |
| 141 | $HNO_2 + FeO^{2+} \rightarrow FeOH^{2+} + NO_2$ | $1.1 \times 10^4$ | 4150 | (Jacobsen et al., 1998) |
| 142 | $NO_2^- + FeO^{2+} \xrightarrow{H^+} FeOH^{2+} + NO_2$ | $1.0 \times 10^5$ | | (Jacobsen et al., 1998) |
| 143 | $HSO_3^- + FeO^{2+} \rightarrow FeOH^{2+} + SO_3^-$ | $2.5 \times 10^5$ | | (Jacobsen et al., 1998) |
| 144 | $HCOOH + FeO^{2+} \xrightarrow{O_2} FeOH^{2+} + CO_2 + HO_2$ | 160 | 2680 | (Jacobsen et al., 1998) |
| 145 | $HCOO^- + FeO^{2+} \xrightarrow{H^+,O_2} FeOH^{2+} + CO_2 + HO_2$ | $3.0 \times 10^5$ | | (Jacobsen et al., 1998) |
| 146 | $FeOH^{2+} + HSO_3^- \rightarrow FeSO_3^+ + H_2O$ | $4.0 \times 10^6$ | | (Lente and Fabian, 2002) |
| 147 | $FeSO_3^+ + H^+ \xrightarrow{OH^-} FeOH^{2+} + HSO_3^-$ | $2.08 \times 10^3$ | | (Lente and Fabian, 2002) |
| 148 | $FeSO_3^+ \rightarrow Fe^{2+} + SO_3^-$ | 0.19 | | (Lente and Fabian, 2002) |
| 149 | $Fe^{2+} + SO_3^- \rightarrow FeSO_3^+$ | $3.0 \times 10^6$ | 5605 | (Buxton et al., 1999) |
| 150 | $FeOH^{2+} + SO_3^- \rightarrow Fe^{2+} + HSO_4^-$ | $3.0 \times 10^5 + 7.6 \times 10^6 \times 1.64 \times 10^{-3}\ [H^+]^{-1}$ | | (Warneck, 2018) |
| 151 | $OH + Fe^{2+} \rightarrow FeOH^{2+}$ | $4.3 \times 10^8$ | 1100 | (Christensen and Sehested, 1981) |
| 152 | $H_2O_2 + FeOH^{2+} \rightarrow HO_2 + H_2O + Fe^{2+}$ | $2 \times 10^{-3}$ | | (Walling and Goosen, 1973) |
| 153 | $O_2^- + FeOH^{2+} \rightarrow O_2 + Fe^{2+} + OH^-$ | $1.5 \times 10^8$ | | (Rush and Bielski, 1985) |
| 154 | $HO_2 + FeOH^{2+} \rightarrow Fe^{2+} + O_2 + H_2O$ | $1.3 \times 10^5$ | | (Ziajka et al., 1994) |
| 155 | $O_2^- + Fe^{2+} \xrightarrow{H^+,H_2O} H_2O_2 + FeOH^{2+}$ | $1.0 \times 10^7$ | | (Rush and Bielski, 1985) |
| 156 | $HO_2 + Fe^{2+} \xrightarrow{H_2O} H_2O_2 + FeOH^{2+}$ | $1.2 \times 10^6$ | 5050 | (Jayson et al., 1973) |
| 157 | $NO_3 + Fe^{2+} \xrightarrow{OH^-} NO_3^- + FeOH^{2+}$ | $8 \times 10^6$ | | (Pikaev et al., 1974) |
| 158 | $FeOH^{2+} + HSO_3^- \rightarrow Fe^{2+} + SO_3^- + H_2O$ | 39 | | (Ziajka et al., 1994) |
| 159 | $Fe^{2+} + SO_5^- \xrightarrow{H_2O} FeOH^{2+} + HSO_5^-$ | $4.3 \times 10^7$ | | (Herrmann et al., 1996) |
| 160 | $Fe^{2+} + HSO_5^- \rightarrow FeOH^{2+} + SO_4^-$ | $3 \times 10^4$ | | (Ziajka et al., 1994) |
| 161 | $Fe^{2+} + SO_4^- \xrightarrow{H_2O} FeOH^{2+} + SO_4^{2-} + H^+$ | $3.5 \times 10^7$ | | (Ziajka et al., 1994) |
| 162 | $Fe^{2+} + S_2O_8^{2-} \xrightarrow{H_2O} FeOH^{2+} + SO_4^{2-} + SO_4^- + H^+$ | 17 | | (Buxton et al., 1997) |
| 163 | $SO_4^- + Fe^{2+} \rightarrow FeSO_4^+$ | $3 \times 10^8$ | | (Mcelroy and Waygood, 1990) |
| 164 | $FeOH^{2+} + SO_4^{2-} \rightarrow FeSO_4^+ + OH^-$ | $3.2 \times 10^3$ | | (Brandt and Vaneldik, 1995) |
| 165 | $FeSO_4^+ \xrightarrow{OH^-} FeOH^{2+} + SO_4^{2-}$ | $1.8 \times 10^5$ | | (Brandt and Vaneldik, 1995) |
| 166 | $Fe^{2+} + O_2 \xrightarrow{OH^-} FeOH^{2+} + O_2^-$ | $8.8 \times 10^{-2}$ | | (Santana-Casiano et al., 2005) |
| 167 | $Fe^{2+} + O_2^- \xrightarrow{OH^-} FeOH^{2+} + O_2^{2-}$ | $10^7$ | | (De Laat and Le, 2005) |
| 168 | $O_2^- + FeSO_4^+ \rightarrow Fe^{2+} + SO_4^{2-} + O_2$ | $1.5 \times 10^8$ | | (Rush and Bielski, 1985) |
| 169 | $HO_2 + FeSO_4^+ \rightarrow Fe^{2+} + SO_4^{2-} + O_2 + H^+$ | $1.0 \times 10^3$ | | (Rush and Bielski, 1985) |
| 170 | $Fe^{2+} + H_2O_2 \rightarrow FeO^{2+} + H_2O$ | See ref. | | (Tong et al., 2017) |

N-chemistry

| No. | Reaction | Rate | E/R | Reference |
|---|---|---|---|---|
| 171 | $NO_2^- \xrightarrow{H^+,h\nu} NO + OH$ | See ref. | | (Zellner et al., 1990) |
| 172 | $NO_3 \xrightarrow{H^+,h\nu} NO_2 + OH$ | See ref. | | (Zellner et al., 1990) |
| 173 | $N_2O_5 + H_2O \rightarrow 2\ H^+ + 2\ NO_3^-$ | $5 \times 10^9$ | | (Herrmann et al., 2000) |
| 174 | $NO_3 + OH^- \rightarrow NO_3^- + OH$ | $9.4 \times 10^7$ | 2700 | (Exner et al., 1992) |
| 175 | $NO_3 + H_2O_2 \rightarrow NO_3^- + H^+ + HO_2$ | $4.9 \times 10^6$ | 2000 | (Herrmann et al., 1994) |
| 176 | $NO_3 + HSO_3^- \rightarrow NO_3^- + H^+ + SO_3^-$ | $1.3 \times 10^9$ | 2000 | (Exner et al., 1992) |
| 177 | $NO_3 + SO_3^{2-} \rightarrow NO_3^- + SO_3^-$ | $3.0 \times 10^8$ | | (Exner et al., 1992) |

| | | | | |
|---|---|---|---|---|
| 178 | $NO_3 + HSO_4^- \rightarrow NO_3^- + H^+ + SO_4^-$ | $2.6 \times 10^5$ | | (Raabe, 1996) |
| 179 | $NO_3 + SO_4^{2-} \rightarrow NO_3^- + SO_4^-$ | $5.6 \times 10^3$ | | (Logager et al., 1993) |
| 180 | $NO_2 + OH \rightarrow NO_3^- + H^+$ | $1.2 \times 10^{10}$ | | (Wagner et al., 1980) |
| 181 | $NO_2 + O_2^- \rightarrow NO_2^- + O_2$ | $1 \times 10^8$ | | (Warneck and Wurzinger, 1988) |
| 182 | $NO_2 + NO_2 \xrightarrow{H_2O} HNO_2 + NO_3^- + H^+$ | $8.4 \times 10^7$ | -2900 | (Park and Lee, 1988) |
| 183 | $O_2NO_2^- \rightarrow NO_2^- + O_2$ | $4.5 \times 10^{-2}$ | | (Lammel et al., 1990) |
| 184 | $NO_2^- + NO_3 \rightarrow NO_3^- + NO_2$ | $1.4 \times 10^9$ | 0 | (Herrmann and Zellner, 1998) |
| 185 | $SO_4^- + NO_3^- \rightarrow SO_4^{2-} + NO_3$ | $5.0 \times 10^4$ | | (Exner et al., 1992) |
| 186 | $HCOOH + NO_3 \xrightarrow{O_2} NO_3^- + H^+ + HO_2 + CO_2$ | $3.8 \times 10^5$ | 3400 | (Exner et al., 1994) |
| 187 | $HCOO^- + NO_3 \xrightarrow{O_2} NO_3^- + HO_2 + CO_2$ | $5.1 \times 10^7$ | 2200 | (Exner et al., 1994) |
| 188 | $NO_2 + HO_2 \rightarrow HO_2NO_2$ | $1.0 \times 10^7$ | | (Warneck and Wurzinger, 1988) |
| 189 | $HO_2NO_2 \rightarrow NO_2 + HO_2$ | $4.6 \times 10^{-3}$ | | (Warneck and Wurzinger, 1988) |
| 190 | $SO_4^- + NO_2^- \rightarrow SO_4^{2-} + NO_2$ | $9.8 \times 10^8$ | 1500 | (Wine et al., 1989) |
| 191 | $NO + NO_2 \xrightarrow{H_2O} 2 NO_2^- + 2 H^+$ | $2.0 \times 10^8$ | 1500 | (Lee, 1984) |
| 192 | $NO + OH \rightarrow NO_2^- + H^+$ | $2.0 \times 10^{10}$ | 1500 | (Strehlow and Wagner, 1982) |
| 193 | $HNO_2 + OH \rightarrow NO_2 + H_2O$ | $1.0 \times 10^9$ | 1500 | (Rettich, 1978) |
| 194 | $NO_2^- + OH \rightarrow NO_2 + OH^-$ | $1.0 \times 10^{10}$ | 1500 | (Treinin and Hayon, 1970) |
| 195 | $HNO_2 + H_2O_2 \xrightarrow{H^+} NO_3^- + 2 H^+ + H_2O$ | $6.3 \times 10^3 [H^+]$ | 6693 | (Lee and Lind, 1986) |
| 196 | $NO_2^- + O_3 \rightarrow NO_3^- + O_2$ | $5.0 \times 10^5$ | 6950 | (Damschen and Martin, 1983) |
| 197 | $NO_3 + HO_2 \rightarrow NO_3^- + H^+ + O_2$ | $4.5 \times 10^9$ | 1500 | (Jacob, 1986) |
| 198 | $NO_3 + O_2^- \rightarrow NO_3^- + O_2$ | $1.0 \times 10^9$ | 1500 | (Jacob, 1986) |
| 199 | $2 NO_2 + HSO_3^- \xrightarrow{H_2O} SO_4^{2-} + 3 H^+ + 2 NO_2^-$ | $2.0 \times 10^6$ | 0 | (Lee and Schwartz, 1982) [c] |
| 200 | $NO_2 + NO_2 \rightarrow N_2O_4$ | $4.5 \times 10^8$ [b] | | (Graedel and Weschler, 1981) |
| 201 | $N_2O_4 \xrightarrow{H_2O} 2 H^+ + NO_2^- + NO_3^-$ | $1.0 \times 10^3$ [b] | | (Graedel and Weschler, 1981) |
| 202 | $NO_3 + H_2O \rightarrow NO_3^- + OH + H^+$ | 6.0 | 4500 | (Rudich et al., 1996) |
| 203 | $NO_3^- + OH + H^+ \rightarrow NO_3 + H_2O$ | $1.4 \times 10^8$ | | (Rudich et al., 1996) |
| 204 | $HSO_5^- + NO_2^- \rightarrow HSO_4^- + NO_3^-$ | 0.31 | 6646 | (Edwards and Mueller., 1962) |
| 205 | $HO_2NO_2 + HSO_3^- \rightarrow HSO_4^- + NO_3$ | $3.5 \times 10^5$ | | (Amels et al., 1996) |
| | | | | |
| | **Carbonate chemistry** | | | |
| 206 | $HCO_3^- + O_2^- \rightarrow HO_2^- + CO_3^-$ | $1.5 \times 10^6$ | 0 | (Schmidt, 1972) |
| 207 | $CO_3^- + H_2O_2 \rightarrow HO_2 + HCO_3^-$ | $8.0 \times 10^5$ | 2820 | (Behar et al., 1970) |
| 208 | $CO_2^- + O_2 \rightarrow O_2^- + CO_2$ | $2.4 \times 10^9$ | | (Tan et al., 2009) |
| 209 | $CO_3^- + O_2^- \rightarrow CO_3^{2-} + O_2$ | $6.8 \times 10^8$ | | (Tan et al., 2009) |
| 210 | $CO_3^- + HCOO^- \rightarrow HCO_3^- + CO_2^-$ | $1.5 \times 10^5$ | | (Tan et al., 2009) |
| 211 | $NO_2^- + CO_3^- \rightarrow CO_3^{2-} + NO_2$ | $6.6 \times 10^5$ | 850 | (Huie et al., 1991) |
| 212 | $HCOO^- + CO_3^- \xrightarrow{O_2} CO_3^{2-} + HO_2 + CO_2$ | $1.4 \times 10^5$ | 3300 | (Zellner et al., 1996) |
| 213 | $HCO_3^- + OH \rightarrow H_2O + CO_3^-$ | $1.7 \times 10^7$ | 1900 | (Exner, 1990) |
| 214 | $CO_3^{2-} + OH \rightarrow OH^- + CO_3^-$ | $3.9 \times 10^8$ | 2840 | (Buxton et al., 1988b; Buxton et al., 1988a) |
| 215 | $CO_3^{2-} + SO_4^- \rightarrow SO_4^{2-} + CO_3^-$ | $4.1 \times 10^7$ | | (Herrmann et al., 2000) |
| 216 | $HCO_3^- + SO_4^- \rightarrow SO_4^{2-} + CO_3^- + H^+$ | $2.8 \times 10^6$ | 2090 | (Huie and Clifton, 1990) |
| 217 | $CO_3^{2-} + NO_3 \rightarrow NO_3^- + CO_3^-$ | $4.1 \times 10^7$ | | (Herrmann et al., 2000) |
| 218 | $CO_3^- + CO_3^- \xrightarrow{O_2} 2 O_2^- + 2 CO_2$ | $2.2 \times 10^6$ | | (Huie and Clifton, 1990) |
| 219 | $CO_3^- + Fe^{2+} \xrightarrow{OH^-} CO_3^{2-} + FeOH^{2+}$ | $2 \times 10^7$ | | (Herrmann et al., 2000) |
| 220 | $CO_3^- + HO_2 \rightarrow HCO_3^- + O_2$ | $6.5 \times 10^8$ | | (Herrmann et al., 2000) |
| 221 | $CO_3^- + HSO_3^- \rightarrow HCO_3^- + SO_3^-$ | $1 \times 10^7$ | | (Herrmann et al., 2000) |

| 222 | $CO_3^- + SO_3^{2-} \rightarrow CO_3^{2-} + SO_3^-$ | $5.0 \times 10^6$ | 470 | (Huie et al., 1991) |

[a] n = reaction order – 1. The units are $s^{-1}$ for first-order reactions and $M^{-1} s^{-1}$ for second-order reactions. Reaction rate constant $k = k_{298} exp(-\frac{E_a}{R}(\frac{1}{T} - \frac{1}{298}))$.

[b] The temperature for k is 293 K.

[c] Referred from https://www.osti.gov/biblio/6567096.

There are four parameters in every pair of gas-aqueous phase transfer equilibria. The two parameters in the transfer from the gas phase to the aqueous phase are the molar mass (g mol$^{-1}$) and mass accommodation coefficients of this species. The other two parameters in the transfer from the aqueous phase to the gas phase are Henry's law constants (M atm$^{-1}$) at 298 K ($K_{H298}$) and "ΔH (J mol$^{-1}$) / R (J mol$^{-1}$ K$^{-1}$)", where ΔH is the enthalpy of dissolution.. The Henry's law constant $K_H$ (M atm$^{-1}$) at any temperature T (K) in Eq. (1) can be calculated by Eq. (2):

$$[C_i] = K_H \cdot P_i \tag{1}$$

$$K_H(T) = K_{H298} \cdot exp\left(-\frac{\Delta H}{R}\left(\frac{1}{T} - \frac{1}{298}\right)\right) \tag{2}$$

where $[C_i]$ and $P_i$ are aqueous-phase and gas-phase concentrations of species i in units of mol $L_{water}^{-1}$ and atm, respectively. On the other hand, the concentration of liquid water is a constant value of 55.6 (i.e., 1000/18) mol L$^{-1}$. The initial concentration of soluble Fe(III) ([$Fe^{3+}$]) is set to 5 μM, which refers to the urban conditions from the literature below (Deguillaume et al., 2005; Mao et al., 2013; Jacob, 2000; Shao et al., 2019; Li et al., 2017; Herrmann et al., 2000; Matthijsen et al., 1995).

To facilitate the calculation of gas-phase and aqueous-phase chemistry simultaneously, the methods used in Jacob (1986) and Liu et al. (2012a) are applied in this study, which convert the units of concentrations and reaction rates in the aqueous phase to the same units as those used in gas-phase chemistry (Jacob, 1986; Liu et al., 2012a):

$$[X_i] = 6.023 \times 10^{20} \cdot LWC \cdot [C_i] \tag{3}$$

where $[X_i]$ and $[C_i]$ are aqueous-phase concentrations of species i in units of molecules cm$_{air}^{-3}$ and mol $L_{water}^{-1}$, respectively, and $6.023 \times 10^{20}$ is the product of Avogadro Constant ($6.023 \times 10^{23}$) and unit conversion factor ($10^{-3}$) between $L_{air}^{-1}$ and cm$_{air}^{-3}$. In this way, the chemical systems of both gas and aqueous phases can be numerically solved without distinction.

### 2.3 Model configuration

Two main simulations are conducted in this study. The first simulation (the Original case) is conducted without any modification of the default CAM4 chemistry, with parameterized aqueous-phase oxidation reactions of $SO_2$ by $H_2O_2$ and $O_3$. In the second simulation (the Improved case), since the $F_{cld}$ is nonzero in most grids, two calculations are performed in a cloudy grid cell. In the cloudy part, the parameterized aqueous-phase reactions mentioned above are replaced by detailed in-cloud aqueous-phase chemistry listed in Tables 1a and 1b coupled with default gas chemistry. In the non-cloudy part, the calculation is similar to the Original case but still without parameterized aqueous-phase reactions. Finally, the concentration in each grid is the average of the cloudy and non-cloudy results weighted by $F_{cld}$.

The timestep used in this study is the default 30 minutes in CESM2. In the Improved case, the lifetime of clouds (i.e., the time between the formation and evaporation of clouds) is set equal to the timestep. At t = 0 of each timestep, all the cloud droplets are assumed to be instantaneously and simultaneously formed according to the cloud-related variables such as LWC, $F_{cld}$ and r, and all the water-soluble species (listed in Table 1a) are dissolved into the cloud droplets according to the effective Henry's law constants. The pH value of each grid cell is calculated by the ionization equilibria of ionizable species (listed in Table 1b) and the dissociation of CCN (including sulfate, nitrate and ammonium), assuming that equilibrium and electroneutrality are continuously maintained. Such pH values can directly influence the formation of aqueous-phase sulfate and nitrate of this timestep. At the same time, a given initial concentration of soluble $Fe^{3+}$ (5 μM) is allocated into each cloud droplet. When t = 30 minutes, all the cloud droplets are assumed to instantaneously evaporate. All the species remaining in the aqueous phase are transferred directly back to the gas phase. Low-volatility species such as ammonium, sulfate and nitrate are released directly back to the atmosphere as inorganic aerosols. Meanwhile, the newly formed sulfate and nitrate will further influence the ionization equilibria and the calculation of pH values in the next step, thus forming a fully-coupled feedback system between pH values and concentrations of sulfate and nitrate.

On the basis of the Improved case, more sensitivity cases are simulated to explore the influences of different factors (e.g., the concentration of soluble Fe and the pH value) on the capacity for $SO_2$ oxidation. The process of all these simulations is the same as that of the Improved case. The detailed description of all the model simulations used in this study is summarized in Table S1 in the supplement.

Finally, all the simulations are running for a 2-year period from $1^{st}$ January 2014 to $31^{st}$ December 2015. The first year is used for model spin-up. In this study, we used "https://svn-ccsm-inputdata.cgd.ucar.edu/trunk/inputdata/input/atm/cam/inic/fv/cami-chem_1990-01-01_0.9x1.25_L30_c080724.nc" as the initial data and boundary conditions to provide the initial values of all the physical variables and concentrations of all the chemical species. The output of the simulation is in the form of a daily mean and is then converted to a monthly or seasonal mean for research needs.

### 2.4 Observations for evaluation of global simulation

For the model evaluation, the observational data used in this study are collected from four monitoring networks. The observations in Europe (EU) are obtained from the European Monitoring and Evaluation Programme (EMEP, https://www.emep.int/, last access: 8 August 2020). The observations in the United States (US) are obtained from the U.S. Environmental Protection Agency Air Quality System (EPA, https://aqs.epa.gov/aqsweb/airdata/download_files.html#Raw, last access: 19 July 2020 ). The observations in China (CN) are obtained from the China National Environmental Monitoring Center (CNEMC, https://quotsoft.net/air/, last access: 22 December 2020). The observations in Japan and South Korea (JK) are obtained from the Acid Deposition Monitoring Network in East Asia (EANET,

, last access: 2 November 2020). The locations of monitoring stations are

shown in Fig. 1. All observational data were collected from 1 January 2015 to 31 December 2015. The monthly averages used

for analysis of the results are calculated from raw daily averages or even hourly averages collected from the measurement

networks above. For convenience of comparison, the units of simulated concentration of $SO_2$ are all converted to the forms in

corresponding observational data.

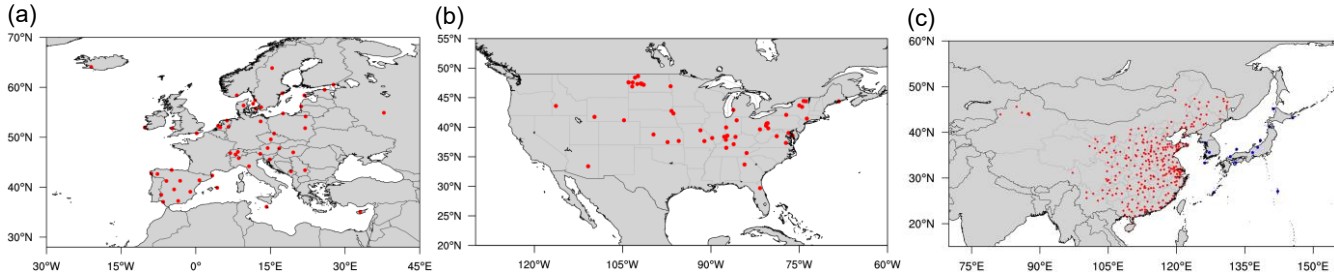

**Figure 1: Locations of monitoring sites from various measurement networks in (a) EU (EMEP), (b) US (EPA) and (c) EA (red points for CNEMC and blue points for EANET).**

## 3    Effects of in-cloud aqueous-phase chemical mechanisms on the simulation of $SO_2$

### 3.1    Simulation of $SO_2$ in the Original case

Figure 2 shows the seasonally averaged surface mixing ratios of $SO_2$ in the Original simulation. There are huge spatial and

temporal differences in the global distribution of $SO_2$. On the one hand, the mixing ratios of $SO_2$ are no more than 0.1 ppbv in

most parts of the world and are basically concentrated in Asia, EU, North America (NA) and South Africa (SA), especially in

Central and East CN. The mixing ratios in NA and EU are mainly in the ranges of 0.1-5 and 1-10 ppbv, respectively. The

mixing ratios in EU are slightly higher than those in NA. Meanwhile, the mixing ratios in the eastern US are evidently higher

than those in the western US. In EA, the mixing ratios in JK range from 0.1 to 5 ppbv throughout the year. The mixing ratios

in most of Central and East CN are in the range of 10-50 ppbv and even higher than 50 ppbv in some regions, which are much

higher than those in other regions. In addition, similar to the US, the mixing ratios in Central and East CN are much higher

than that in Western CN. Such distributions are directly related to the high emissions of $SO_2$ in these regions of CN (Jo et al.,

2020; Xie et al., 2016; Geng et al., 2019).

On the other hand, the mixing ratios of $SO_2$ are remarkably different in the four seasons. They are highest in winter, followed

by spring and autumn, and lowest in summer, especially in Asia and NA. Such seasonal differences are related to both

emissions and the capacity for $SO_2$ oxidation in the gas phase. In winter, due to the increase in heating demand, the combustion

of fossil fuels such as coal could significantly increase the emissions of $SO_2$ (Jo et al., 2020; Xie et al., 2016; Geng et al., 2019;

Feng et al., 2020). At the same time, higher temperatures and stronger sunlight could enhance the gas-phase oxidation of $SO_2$

in summer, which is the opposite in winter. Such phenomena are consistent with multiple studies (Alexander et al., 2009; Huang et al., 2014; Tilgner et al., 2013; Shao et al., 2019).

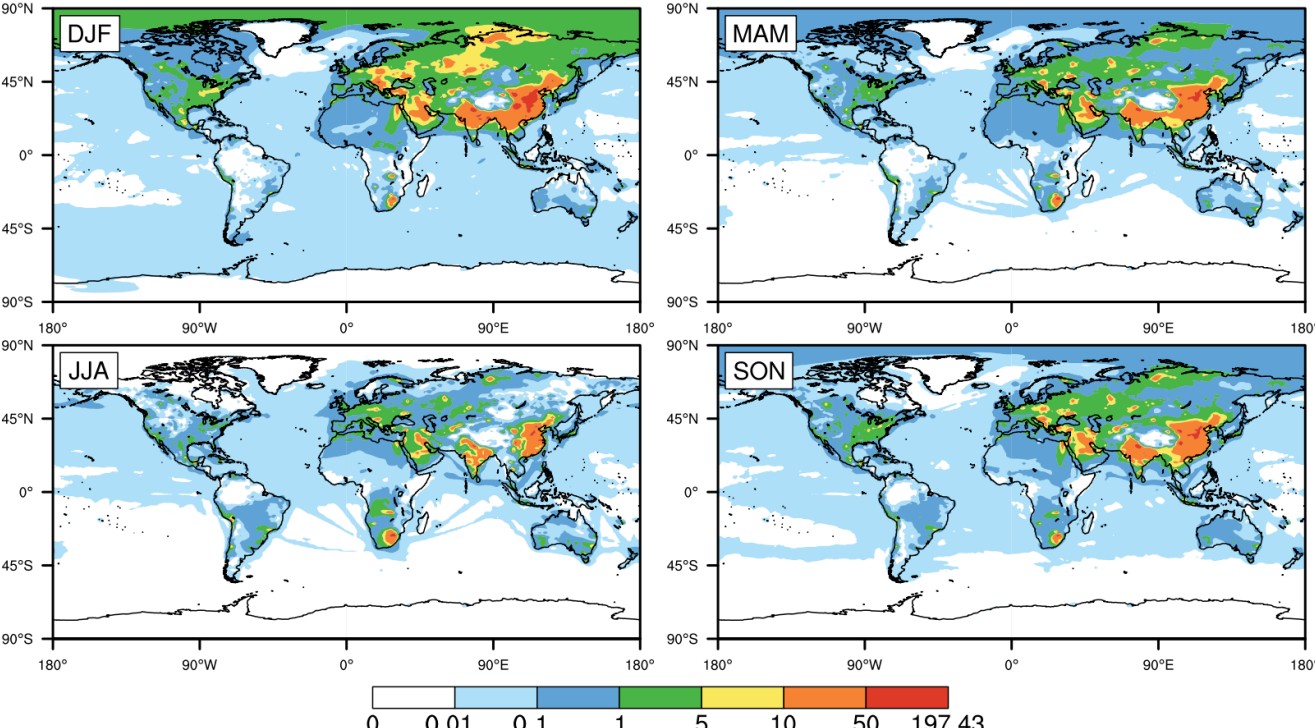

**Figure 2: Global distribution of seasonally averaged surface mixing ratios of SO₂ (unit: ppbv) in 2015, simulated by CESM2 with standard configuration (i.e., the Original case). DJF, MAM, JJA and SON represent December-January-February, March-April-May, June-July-August and September-October-November, respectively, the same below.**

### 3.2 Differences between the Original and Improved simulations

After replacement of default parameterized aqueous-phase reactions with detailed in-cloud aqueous-phase chemistry, the surface mixing ratios of SO₂ generally decrease markedly, as shown in Fig. 3. The extent of the reduction is distinct in different regions and seasons. In general, reductions in SO₂ mainly occur in Asia, EU, NA and SA. The mixing ratios of SO₂ decrease the most in CN, followed by EU, and the least in NA and JK. These results are partly due to the relatively high background mixing ratios in these regions in the Original simulation. Therefore, all the distribution patterns above are also similar to those in the Original simulation. The reductions in SO₂ also differs in various seasons. In NA and EU, the mixing ratios of SO₂ in most regions are reduced by 0.1-5 and 1-10 ppbv in winter, respectively. In spring and autumn, the mixing ratios mainly decrease by 0.1-5 ppbv, which is slightly less than that in winter. However, the reduction in SO₂ in summer is very limited. Notably, the mixing ratios in some areas even increase slightly, which is partly due to the replacement of default parameterized aqueous-phase reactions. Sometimes these simplified and parameterized reactions are even stronger than detailed radical reactions, especially in summer. Similar to Fig. 2, Figure 3 shows that the decline in SO₂ mixing ratios in the eastern US is larger than that in the western US, which is also related to the background mixing ratios in the Original case. However, the situation is different in EA. The mixing ratios decrease significantly in all seasons in Central and East CN. The reduction is

 always more than 1 ppbv, sometimes even greater than 10 ppbv. Again, the reductions in Central and East CN are higher than that in Western CN. These results are also partly related to the background mixing ratios. The decrease in JK ranges from 0.1 to 5 ppbv throughout the year, without obvious fluctuation.

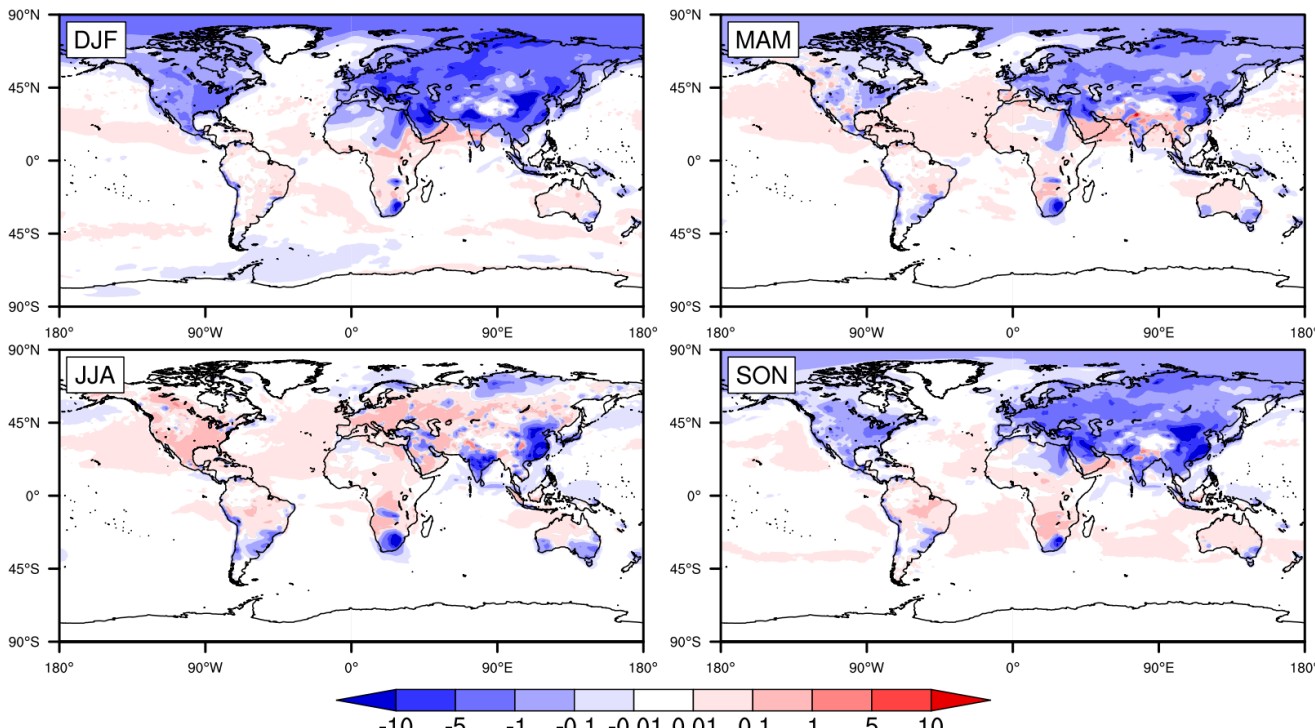

**Figure 3: The differences in global seasonally averaged surface SO₂ mixing ratios between the Improved case and the Original case**
**in 2015 after the incorporation of detailed in-cloud aqueous-phase chemical mechanisms (unit: ppbv).**

In regard to the relative differences between the Original and Improved cases, the results seem slightly different, as shown in Fig. S1 (see the Supplement). Although the reduction sequence is winter > autumn > spring > summer in general, which is very similar to the results above, the regional differences are no longer distinct. In winter, the mixing ratios of SO₂ decrease more than 50% in most regions of EU and NA but no more than 50% in Central and East CN. In contrast, the reductions are 295 very small in EU, NA and JK in summer. However, the decreases exceed 10% in CN and even 50% in some regions (e.g., Shanxi, Hebei, Zhejiang and Fujian Provinces).

Such enhancement of the oxidation capacity can also be reflected in the net chemical loss rate of SO₂. Fig. S2 shows the ratio of the net chemical loss rates between the Improved and Original simulations. The net chemical loss rate increases in most parts of the world (ratios > 1). The seasonal differences in the ratios are winter > autumn > spring > summer, which is still 300 similar to the results above. The ratios in NA and JK are all more than 20 times greater and even above 100 times higher in some regions. Those in EU are more than 20 times greater in winter but less than 10 times higher in summer. The multiples in Western CN are all more than 20 times greater and even more than 100 times greater in some regions, which are much higher than those in East CN, which are only less than 10 times greater.

### 3.3 Comparison between the simulated and observed SO₂ concentrations

The regional annual average mixing ratios between the simulated and observed $SO_2$ are summarized in Table 2. At the same time, scatter plots of $SO_2$ over various sites in the four monitoring networks are also shown in Fig. S3. Clearly, the effect of detailed aqueous-phase chemistry on the improvement in $SO_2$ simulation is remarkable. The annual average mixing ratios in the Original case are 4.3, 1.5, 2.0, and 1.6 times overestimated in EU, US, CN and JK, respectively. After incorporating the detailed aqueous-phase chemistry, these values are reduced by 46%, 41%, 22% and 43%, respectively. The slopes of the linear fitting lines are all close to or even approximately equal to 1 in EU, US and JK.

**Table 2.** **Comparison of regional annual average values among the Observed, Original-simulated and Improved-simulated SO₂ mixing ratios (ppbv) in EU, US, CN and JK in 2015. The observed mixing ratios are calculated by averaging the data from all monitoring stations of various measurement networks. The simulated mixing ratios are calculated by averaging the data from all the grids where the monitoring stations are located.**

| Region | Monitoring network | Obs | Ori | Imp |
|--------|--------------------|-----|-----|-----|
| EU | EMEP [a] | 0.38 | 2.0 | 1.0 |
| US | EPA | 1.1 | 2.7 | 1.6 |
| CN | CNEMC [a] | 10 | 30 | 23 |
| JK | EANET | 0.54 | 1.4 | 0.78 |

[a] The original units of EMEP and CNEMC are μg S m⁻³ and μg m⁻³, respectively. To facilitate comparison, these two units are converted to ppbv. Units conversion: $1\ mol\ mol_{air}^{-1} = 1 \times 10^9\ ppbv = \frac{64 \times 10^6\ P_{air}}{R\,T_{air}}\ \mu g\ m^{-3} = \frac{32 \times 10^6\ P_{air}}{R\,T_{air}}\ \mu g\ S\ m^{-3}$. 64 and 32 are molar masses of $SO_2$ and S, respectively. $10^6$ is the unit conversion coefficient between "g" and "μg". R = 8.314 J mol⁻¹ K⁻¹ is the ideal gas constant. $T_{air}$ (288 K used here) is atmospheric temperature in Kelvin. $P_{air}$ (1.013 × 10⁵ Pa used here) is atmospheric pressure in Pa. The same below.

The comparison between the simulated and observed monthly average mixing ratios of $SO_2$ in the four monitoring networks is shown in Fig. 4. The relative differences between the Original and Improved simulations are also shown in Fig. S4. According to these two figures, compared with the observations, the Original simulation is generally overestimated in all regions, especially in winter. Coupling the detailed in-cloud aqueous-phase chemical mechanisms greatly improves the simulation of $SO_2$. In EU, aqueous-phase reactions significantly improve the simulation of $SO_2$ from October to February. The simulated mixing ratios decrease by more than 60% from the Original case to the Improved case and even by more than 75% in December. The Improved mixing ratios for six months are within the standard deviation of observations. The results in US are even better than those in EU. The mixing ratios of $SO_2$ decrease more in winter, spring and autumn (-30 to -70%) than in summer. All Improved mixing ratios are within the standard deviation of observations. Although the absolute reduction in $SO_2$ over CN is the greatest, the relative improvement in CN is the least due to the excessively high Original mixing ratios of $SO_2$. None of the simulated mixing ratios decrease by more than 40%. None of the Improved mixing ratios are within the standard deviation of observations. The aqueous-phase reactions also greatly improve the simulation in JK. The relative differences in

the four seasons are all close (approximately -30 to -60%). Almost all the Improved mixing ratios are also within the standard deviation of observations.

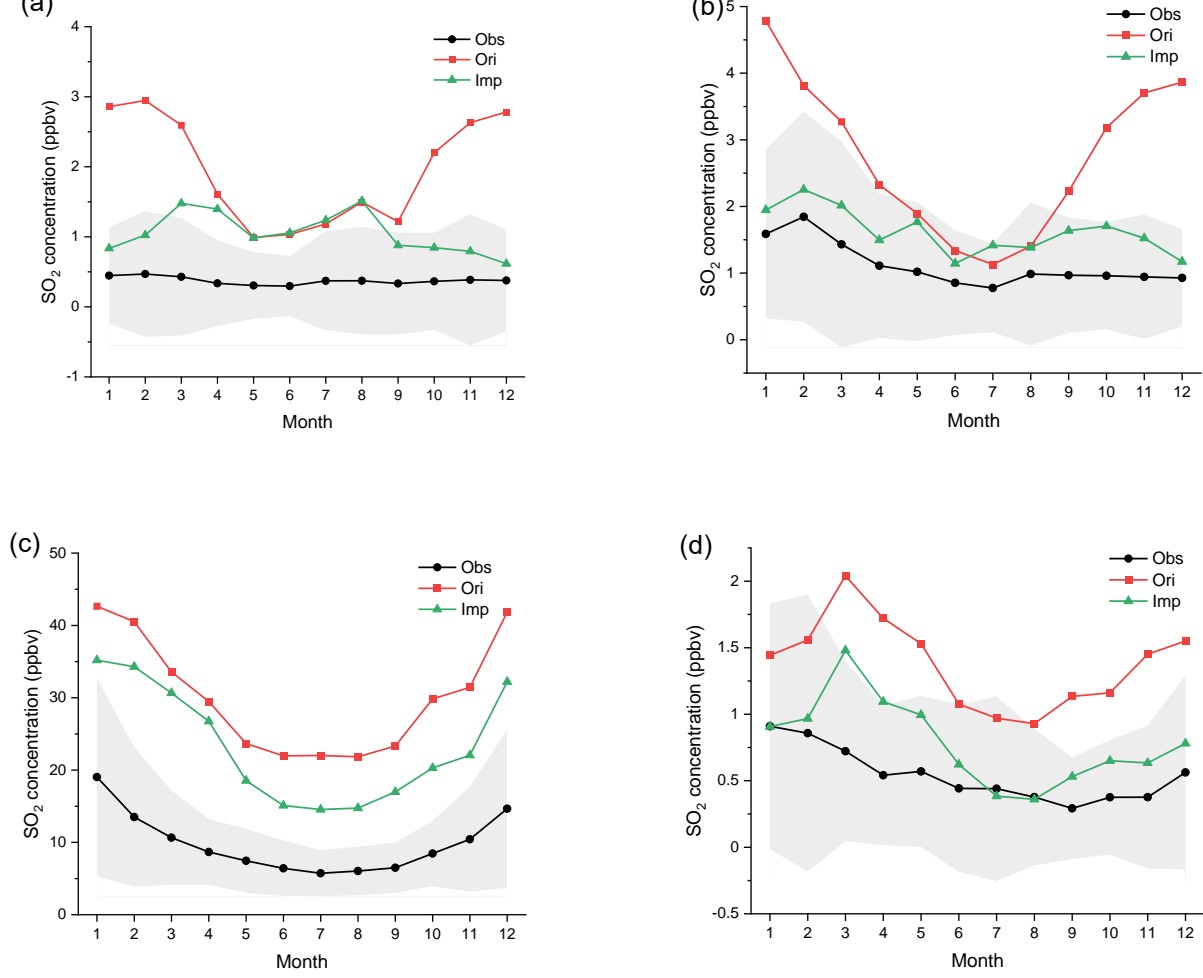

**Figure 4: Regional monthly average mixing ratios (ppbv) of SO₂ in EU, US, CN and JK in 2015. The black, red and green lines represent the Observed, Original-simulated and Improved-simulated mixing ratios, respectively. The gray areas represent the standard deviation of Observed mixing ratios. The corresponding monitoring networks are (a) EMEP, (b) EPA, (c) CNEMC and (d) EANET.**

Overall, the overestimation in winter is more serious than that in summer. At the same time, the improvement from adding aqueous-phase chemistry is much greater in winter than in summer, especially in EU and US. These results indicate the importance of incorporating detailed aqueous-phase chemistry in winter and are highly consistent with the results of some existing studies (Shao et al., 2019; Ma et al., 2018; Huang et al., 2019). Such seasonal differences may be related to the ambient temperature, humidity, and especially sunlight. In summer, both the temperature and sunlight are sufficient to generate a high concentration of ·OH (Lakey et al., 2016). Therefore, gas-phase oxidation is strong and dominant (Cheng et al., 2016). However, due to the weak sunlight in winter, the gas-phase concentration of ·OH is two or three orders of magnitude less than that in summer. In addition, the rate constant is also less than 1/3-1/2 of that in summer owing to the decrease in temperature.

Therefore, the gas-phase photochemical oxidation of $SO_2$ induced by $\cdot OH$ is sharply weakened. These changes indicate the greatly increased importance of aqueous-phase reactions (Elser et al., 2016; Ervens, 2015; Harris et al., 2013; Huang et al., 2018). At the same time, higher humidity and more cloud coverage can provide a more sufficient aqueous environment, which is also beneficial to improve the performance of aqueous-phase reactions, such as those that occur during winter in EU and US and summer in EA.

**4    Contributions of different aqueous-phase chemical mechanisms to the oxidation of $SO_2$**

On the basis of above analysis of the overall detailed aqueous-phase chemistry, it is necessary to discuss the contributions of different aqueous-phase chemical mechanisms in detail. Cases for four different mechanisms are performed with the corresponding reactions in Table 1. See Table S1 for details about the configuration of individual cases. Given the fact that the $HO_x$ chemistry involves most of the critical radicals in aqueous-phase chemistry, the cases of Fe, N and carbonate chemistry

also include the $HO_x$ chemistry. The individual contribution of Fe, N or carbonate chemistry is compared with the $HO_x$-chem alone case.

Figure 5 shows the effects of $HO_x$-chemistry, Fe-chemistry, N-chemistry and carbonate chemistry on surface $SO_2$ (case 3~6 – case 1). Remarkable differences are clearly seen among these four mechanisms. On the one hand, generally speaking, the contributions from both $HO_x$-chemistry and Fe-chemistry to the oxidation of $SO_2$ are significant. Nonetheless, the seasonal

and regional distribution properties of these two chemical mechanisms are obviously different. For $HO_x$-chemistry, the mixing ratios of $SO_2$ decrease in most parts of the world, and the seasonal differences are very small. The reductions generally range from 0.01-0.1 ppbv over the ocean and 0.1-5 ppbv over land. In regard to Fe-chemistry, however, the reduction in $SO_2$ is mostly concentrated on land only, especially in the Northern Hemisphere. The seasonal properties of the reductions are nearly the same as those described in Sect. 3.2. On the other hand, these two chemical mechanisms contribute much more than N-

chemistry to the oxidation of $SO_2$. The decrease in $SO_2$ exceeds 1 ppbv in many regions of Asia, EU and NA due to the effects of Fe-chemistry or $HO_x$-chemistry. Meanwhile, the contribution of N-chemistry almost never exceeds 1 ppbv. Such great disparity may be related to the level of Fe concentrations and pH values in cloud water, which are discussed in Sect. 5. In regard to carbonate chemistry, however, it is difficult to see consistent change in either spatial or temporal $SO_2$ distribution. The mixing ratios of $SO_2$ decreases in some places and seasons but increases in other places and seasons. Moreover, all the

changes are very small, within $\pm 0.1$ ppbv. Therefore, carbonate chemistry has no significant effect on the oxidation of $SO_2$.

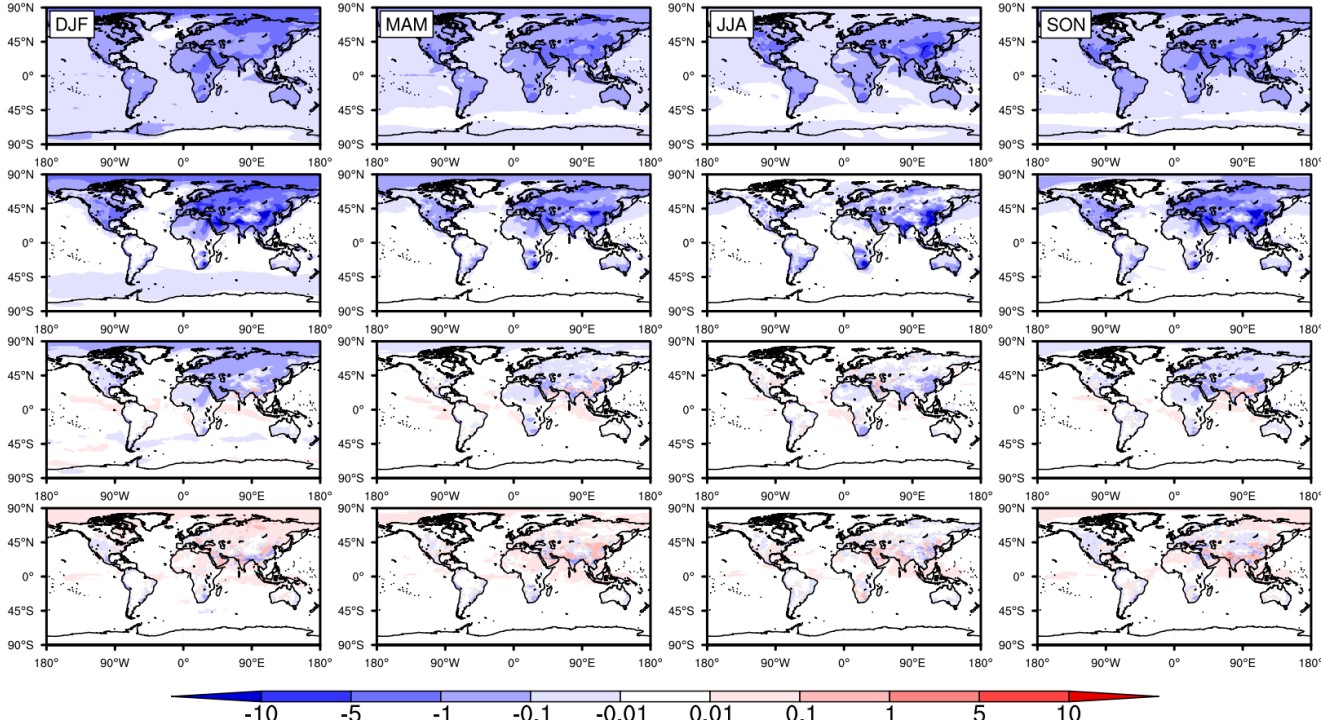

Figure 5: The differences in global seasonally averaged surface $SO_2$ mixing ratios (unit: ppbv) in 2015 with the incorporation of $HO_x$-chemistry, Fe-chemistry, N-chemistry and carbonate chemistry individually from top to bottom, respectively.

The contributions of the different chemical mechanisms discussed above can also be seen from the relative differences, as shown in Fig. S5. $HO_x$-chemistry contributes the most over the ocean in the Southern Hemisphere. At the same time, Fe-chemistry contributes the most over land in the Northern Hemisphere. The mixing ratios of $SO_2$ decrease by more than 50% by both mechanisms. Furthermore, note that although the contribution of carbonate chemistry is quite small, there is an evident decrease over the ocean in the Southern Hemisphere.

**5   Factors affecting the capacity for $SO_2$ oxidation from aqueous-phase reactions**

**5.1   The concentration of soluble Fe**

The concentrations of soluble $[Fe^{3+}]$ are all set to 5 μM in the Improved case. Nevertheless, $[Fe^{3+}]$ varies greatly in different regions, seasons and ambient conditions. For instance, $[Fe^{3+}]$ is generally no more than 0.1 μM under marine conditions and no more than 1 μM under remote continental conditions (Herrmann et al., 2000; Matthijsen et al., 1995; Deguillaume et al.,

2005; Mao et al., 2013; Jacob, 2000; Shao et al., 2019; Li et al., 2017). In many polluted cities, $[Fe^{3+}]$ is much higher than that in remote regions, usually ranging from 5-20 μM and sometimes even exceeding 100 μM (Matthijsen et al., 1995; Deguillaume et al., 2005; Herrmann et al., 2000; Mao et al., 2013; Jacob, 2000; Li et al., 2017). Therefore, in this study, four other levels of initial $[Fe^{3+}]$ (0.1, 1, 20 and 100 μM) are tested with the whole in-cloud aqueous-phase reactions to evaluate the influence of soluble Fe concentration on the capacity for $SO_2$ oxidation. The processing $[Fe^{3+}]$ in these sensitivity cases is identical to the

Improved case except the differences in $Fe^{3+}$ concentrations. All the levels of $[Fe^{3+}]$ are based on the reported values above. See Table S1 for details.

The regional monthly average mixing ratios of $SO_2$ in four regions are shown in Fig. 6. In all four regions, the simulated $SO_2$ first still increase in summer when initial $[Fe^{3+}]$ is 0.1 μM and then decrease considerably when initial $[Fe^{3+}]$ increases from 0.1 μM to 5 μM but decline only slightly when initial $[Fe^{3+}]$ increases to 20 μM. The two lines of "[Fe] = 20 μM" and "[Fe] =

100 μM" almost overlap and cannot be distinguished clearly in EU, US and JK. Only in CN does the mixing ratios of $SO_2$ further decrease obviously when initial $[Fe^{3+}]$ increases from 5 μM to 100 μM. There are many steel and coal factories and power plants in CN. These results imply that there may be a strong correlation between high emissions of $SO_2$ and iron and that the concentrations of Fe in CN may be higher than those in other regions.

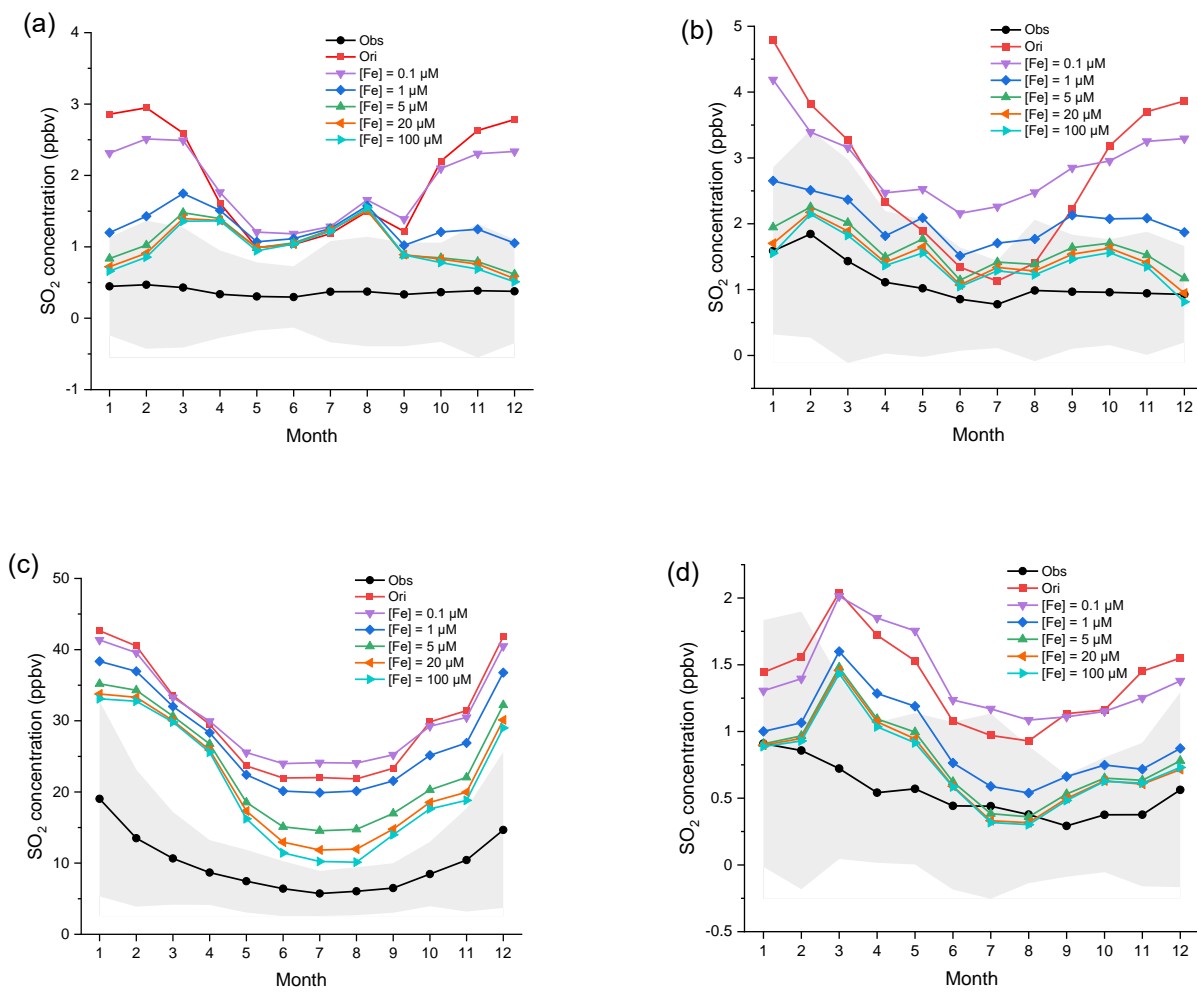

**Figure 6: Regional monthly average mixing ratios (ppbv) of $SO_2$ in EU, US, CN and JK in 2015. The black and red lines represent the Observed and Original-simulated mixing ratios of $SO_2$, respectively. Other lines represent $SO_2$ mixing ratios with different soluble $[Fe^{3+}]$. $[Fe^{3+}]$ from top to bottom are 0.1, 1, 5 (i.e., the Improved case), 20 and 100 μM, respectively. The gray areas represent the standard deviation of Observed mixing ratios. The corresponding monitoring networks are (a) EMEP, (b) EPA, (c) CNEMC and**

**(d) EANET.**

Such an effect on the capacity for $SO_2$ oxidation by $[Fe^{3+}]$ chemistry can also be seen in Fig. S6. The capacity for $SO_2$ oxidation is enhanced with increasing $[Fe^{3+}]$ on the whole. When initial $[Fe^{3+}]$ is only 0.1 μM, the effect of Fe-chemistry is still quite weak. The effect is rapidly enhanced when initial $[Fe^{3+}]$ increases from 0.1 μM to 5 μM. However, such enhancement becomes markedly less when initial $[Fe^{3+}]$ is greater than 20 μM. The mixing ratios of $SO_2$ is almost unchanged when initial $[Fe^{3+}]$

increases to 100 μM. This result indicates that the effect of increasing $[Fe^{3+}]$ on the capacity for $SO_2$ oxidation has a threshold. Too much $[Fe^{3+}]$ will not further facilitate the oxidation of $SO_2$. The reason for such a limitation is discussed below.

In any case, a higher concentration of soluble Fe results in an improvement in the $SO_2$ simulation compared to the observations.

## 5.2    The pH value

As mentioned in the Introduction, the pH value in cloud water is a key parameter for aqueous-phase chemistry, which could

directly affect ionization equilibria and gas-aqueous mass transfer processes. There are expressions for the rate constants of several aqueous-phase reactions, and some expressions include pH values directly. Therefore, the pH value could affect the various aqueous-phase reaction rates, especially that of N-chemistry (Shao et al., 2019; Li et al., 2017; Cheng et al., 2016; He et al., 2018; He and He, 2020). Therefore, it is necessary to discuss the influence of the variation of pH value on the capacity for $SO_2$ oxidation. In this study, there are four sets of pH values (i.e., 3, 4, 5 and 6) prescribed in the following sensitivity tests

(Table S1). All the pH values are referenced from previous studies (Herrmann et al., 2000; Matthijsen et al., 1995; Shao et al., 2019; Guo et al., 2017; Cheng et al., 2016). $[Fe^{3+}]$ is set to 5 μM. However, it is difficult to see obvious differences among these four pH levels in all seasons. Only a small decrease in $SO_2$ can be seen in most regions from pH 3 to 4. The reduction in $SO_2$ is almost the same from pH 4 to 6. This result indicates that the effect of increasing the pH value on the capacity for $SO_2$ oxidation is limited.

The global distributions of $SO_2$ in different seasons shown in Fig. S7 have similar features. Although the capacity for $SO_2$ oxidation increases to some extent from pH 3 to 4 in all four regions, the changes from pH 4 to 6 are very small.

Notwithstanding, a higher pH value doubtless enhances the capacity for $SO_2$ oxidation and results in simulated values that are closer to the observations, which is similar to the influence of the soluble Fe concentration (Shi et al., 2019; Shao et al., 2019; Li et al., 2017; Cheng et al., 2016).

It seems that the influence of the pH value on the aqueous-phase chemistry is much weaker than that of the soluble Fe concentration. When further discussing the effect of the pH value on N-chemistry, $HO_x$-chemistry or Fe-chemistry individually, however, the situation is quite different, as shown in Table S1 and Figs. S8-10. When the pH increases from 3 to 6, the capacity for $SO_2$ oxidation from N-chemistry and $HO_x$-chemistry is evidently enhanced at all times. When the pH is 6, the oxidation capacity from N-chemistry and $HO_x$-chemistry becomes almost as strong as that from Fe-chemistry with a high concentration

of soluble Fe. This indicates that the capacity for $SO_2$ oxidation from N-chemistry and $HO_x$-chemistry is greatly affected by the pH value (Wang et al., 2020; Cheng et al., 2016; He et al., 2018; Li et al., 2018b; He and He, 2020). In contrast, the capacity

for $SO_2$ oxidation from Fe-chemistry is the opposite. When $[Fe^{3+}]$ is set at the default medium level (5 μM), regardless of the pH, there are no remarkable changes in $SO_2$ mixing ratios, and the capacity for $SO_2$ oxidation from Fe-chemistry is nearly the same, especially when the pH ranges from 4 to 6. This indicates that the Fe-chemistry is not significantly affected by pH.

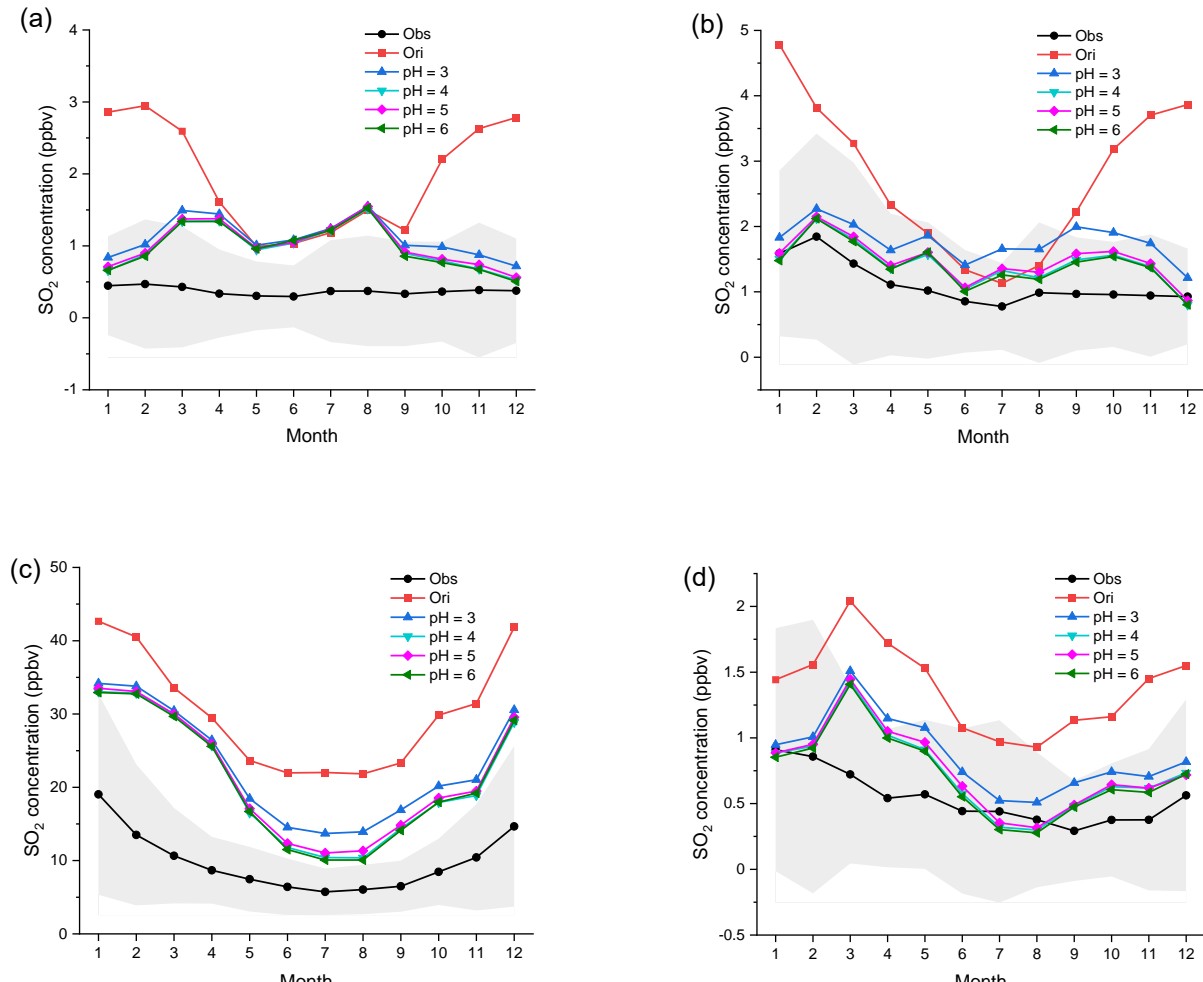

**Figure 7: Regional monthly average mixing ratios (ppbv) of $SO_2$ in EU, US, CN and JK in 2015. The black and red lines represent the Observed and Original-simulated mixing ratios of $SO_2$, respectively. Other lines represent $SO_2$ mixing ratios at different pH values. The pH values from top to bottom are 3, 4, 5 and 6, respectively. $[Fe^{3+}]$ is set to 5 μM. The gray areas represent the standard**
**deviation of Observed mixing ratios. The corresponding monitoring networks are (a) EMEP, (b) EPA, (c) CNEMC and (d) EANET.**

These results well explain why the contribution of N-chemistry is much smaller than those of Fe-chemistry and $HO_x$-chemistry in Sect. 4. According to the simulation, the pH value in cloud water is generally in the range of 3-5. This pH range is highly consistent with those in previous studies (Herrmann et al., 2000; Matthijsen et al., 1995; Shao et al., 2019; Guo et al., 2017; Cheng et al., 2016). As seen in Fig. 5, the capacity for $SO_2$ oxidation from N-chemistry is between pH 4 and 5, which is still
not strong enough. Consequently, the capacity for $SO_2$ oxidation from N-chemistry is largely limited by the relatively low pH values in cloud water.

As analyzed in the sections above, it is worth noting that regardless of the high soluble Fe concentration or high pH value for different chemical mechanisms, the reduction in $SO_2$ always seems to reach a very similar limitation, whose global distribution

and regional monthly average mixing ratios are also almost the same. This is not only related to soluble Fe concentration, pH value or the chemical properties of various mechanisms themselves but also derived from the exhaustion of $SO_2(aq)$ by detailed aqueous-phase chemistry in a finite cloud. The aqueous-phase chemistry cannot affect regions without clouds because the total $SO_2$ is calculated by weighted averages of "cloudy" and "non-cloudy" conditions according to $F_{cld}$. The overestimated $SO_2$ is sometimes caused by a shortage of clouds, especially in CN. Therefore, only more cloud coverage or lower emissions may further reduce the overestimation.

Consequently, it is easy to conclude that the oxidation capacity of Fe-chemistry and $HO_x$-chemistry is much higher than that of N-chemistry when the pH is less than 5, but evaluating their relative importance at high pH is difficult. The cloud content and substrate concentration become the limiting factors. Therefore, a comprehensive investigation of cloud pH in different seasons and different places is urgently needed.

## 5.3 Discussion and uncertainty analysis

Recent studies show that hydroxymethanesulfonate (HMS), formed by aqueous-phase reactions of dissolved HCHO and $SO_2$, is an abundant organosulfur compound in aerosols during winter haze episodes, and suggest that aqueous clouds act as the major medium for HMS chemistry (Moch et al., 2020; Song et al., 2021). Therefore, it's necessary to further investigate the influence of this organic chemistry on the in-cloud aqueous-phase chemistry system in CESM2. We tried to incorporate 10 aqueous-phase organic species and 60 related reactions, including the reactions related to $CH_3OH$, HCHO, $CH_3OOH$ and HMS, as shown in Tables S2a and S2b. We conducted additional simulations for testing the contribution from this organic chemistry. As shown in Fig. S11, incorporating this organic chemistry has a minor effect on $SO_2$ concentrations, similar to that of carbonate chemistry.

In addition to the soluble Fe concentration and pH value discussed above, there are some other factors that may also affect the capacity for $SO_2$ oxidation and increase the uncertainty of the simulation. First, the simulation of variables related to cloud properties (such as LWC, $F_{cld}$ and r) directly determines the contribution of aqueous-phase chemistry. However, the simulation of these variables is also one of the greatest uncertainties (Zhang et al., 2019; Faloona, 2009). In addition, the initial valence of soluble Fe and the proportion of various valences are related to the capacity of Fe-chemistry. The higher the proportion of $Fe^{3+}$ is, the stronger the atmospheric oxidizability and the more helpful for the oxidation of $SO_2$ (Jacob, 2000; Deguillaume et al., 2005; Huang et al., 2014; Alexander et al., 2009). Moreover, the emissions and solubility of Fe vary greatly in different regions. For instance, the total concentration of atmospheric Fe is generally measured in the range of 1-1000 ng $m_{air}^{-3}$, and the solubility of elemental Fe varies from less than 1% to 10% (Cwiertny et al., 2008; Hsu et al., 2010; Sedwick et al., 2007; Sholkovitz et al., 2009; Hsu et al., 2013; Heal et al., 2005; Ingall et al., 2018; Mao et al., 2013; Itahashi et al., 2018; Shelley et al., 2018; Mcdaniel et al., 2019; Conway et al., 2019; Shi et al., 2020; Myriokefalitakis et al., 2018; Wang et al., 2015). Meanwhile, the simulated LWC usually ranges from $10^{-8}$ to $10^{-5}$ $L_{water}$ $L_{air}^{-1}$ in CESM2 and other model studies (Herrmann et

al., 2000; Jacob, 1986; Matthijsen et al., 1995; Liu et al., 2012a; Herrmann et al., 2015). In addition, $F_{cld}$ should also be considered. Therefore, the concentration of soluble Fe can be calculated in a range from less than $10^{-3}$ μM to $10^3$ μM, involving great uncertainties. It is also the reason why the dust aerosol is simulated but not coupled with soluble Fe in this study. At the same time, the proportions of aerosols containing sulfate, nitrate and ammonium in the aqueous phase could directly affect the pH of cloud water. The simulated pH value of cloud water itself is one of the sources of uncertainty (Shi et al., 2019; Xue et al., 2016). Finally, some sources of kinetic parameters for the aqueous-phase reactions are outdated. They may also not be accurate enough because measurement conditions in the laboratory are different from the conditions of the real atmosphere. These issues influence the accuracy of the reaction rates and increase the uncertainty of the simulation.

In addition, there are factors that affect the performance of the simulation to a certain degree. First, an accurate emission inventory is the premise for improving the simulation (Im et al., 2018; De Meij et al., 2006; Liu et al., 2018; Buchard et al., 2014). The data sources and resolutions of various emission types could affect the reliability of the inventory. For instance, regardless of how the parameters discussed above are optimized, the concentration of $SO_2$ in CN is always overestimated, which may be related to the uncertainties in emission inventories. The emissions of $SO_2$ in CN have decreased considerably in recent years, which may lead to biases in the simulations (Jo et al., 2020; Xie et al., 2016; Geng et al., 2019). Meanwhile, the meteorological data include information on the water content, wind, temperature and pressure, which all influence the formation and movements of clouds. Therefore, the reliability of meteorological data is also related to the uncertainty of simulations with in-cloud chemistry (Bei et al., 2017; Liu et al., 2007). At the same time, the simulation of the $SO_2$ wet deposition process also involves great uncertainty. Furthermore, the selection of monitoring stations determines the quality of observational data. As a global model, CESM2 used in this study has a resolution that is still not fine enough to accurately simulate regions that are too remote or too close to pollution sources. The simulation of each grid can represent only the average level of a region. Therefore, the monitoring stations should also represent the average level of the region. Otherwise, the limitation of the model resolution also increases the deviation of the comparison with observations and the uncertainty of the simulations. Finally, there are slight numerical fluctuations during the calculation of the model itself, but the uncertainty from the fluctuations is very small and can be ignored, especially after the results are averaged.

## 6 Conclusion

To improve the global simulation of $SO_2$, in this study, we used CESM2 to evaluate the effects of detailed in-cloud aqueous-phase reaction mechanisms on the capacity for $SO_2$ oxidation. After the replacement of default simplified and parameterized aqueous-phase reactions with detailed in-cloud aqueous-phase reactions, the overestimation of surface $SO_2$ generally decreases significantly. The reductions vary in different regions and seasons. Most them are in the range of 0.1-10 ppbv and some can be greater than 10 ppbv in some regions of CN. The net chemical loss rate of $SO_2$ also increases substantially. When compared

with the observations, the simulated values that incorporate detailed aqueous-phase chemistry improve greatly, making the simulations much closer to the observations. The biases of annual average simulated mixing ratios decrease by 46%, 41%, 22% and 43% in EU, US, CN and JK, respectively. The mixing ratio even decreases by approximately 70% in winter in EU, which is very close to the observed value. The mixing ratios of $SO_2$ in CN are still highly overestimated, although they decrease considerably. Aqueous-phase chemistry contributes more in EU, US and JK than in CN, which may be related to cloud coverage and emissions.

The contribution of each aqueous-phase mechanism to the simulation of $SO_2$ also differs significantly. Fe-chemistry and $HO_x$-chemistry contribute more to the capacity for $SO_2$ oxidation than N-chemistry. Carbonate chemistry has no significant effect on the oxidation of $SO_2$. Several factors could influence the capacity for $SO_2$ oxidation. Higher concentrations of soluble Fe and higher pH values could further enhance the oxidation capacity and improve the simulation of $SO_2$. In addition, the oxidation capacities from N-chemistry and $HO_x$-chemistry are strongly affected by pH values and increase rapidly with increasing pH. The oxidation capacity from Fe-chemistry is almost unaffected by pH. Many other factors also affect the aqueous-phase chemistry and the simulation of $SO_2$. Regardless of which factor changes, there is still a limitation on the improvement in the simulations because of limited cloud coverage in the aqueous phase.

This study emphasizes the importance of aqueous-phase chemical mechanisms for $SO_2$ oxidation. These mechanisms are helpful to improve the simulation of $SO_2$ by CESM2, deepening the understanding of $SO_2$ oxidation and the formation of sulfate, $PM_{2.5}$ and even haze days. A better simulation of $SO_2$ is a prerequisite for better representing sulfate, which further influences cloud microphysics, radiation transfer and climate change.

However, some aspects still need to be further studied and improved in the future. For instance, there is a high degree of uncertainty in the concentration of soluble Fe owing to the dramatically large variation in the total atmospheric Fe content and Fe solubility in different regions. At the same time, there are few observational data or emission inventories of soluble Fe. Therefore, the contribution of Fe-chemistry to the capacity for $SO_2$ oxidation is uncertain under different atmospheric conditions and difficult to evaluate accurately. Meanwhile, many variables and parameters related to the simulated clouds are also uncertain, such as LWC, $F_{cld}$, r, pH values in clouds, wet deposition processes, and proportions of inorganic aerosols in the aqueous phase. Therefore, it is urgently necessary to compare these variables with observational data if possible. Moreover, the effect of aqueous-phase chemistry on $SO_2$ at high altitude is not discussed in this study. These issues will be examined in our future work.

**Code availability**

The Community Earth System Model 2(CESM2) developed by the National Center for Atmospheric Research can be downloaded online (https://www.cesm.ucar.edu/models/cesm2/). All codes used to generate the results of this study are available from the authors upon request.

**Data availability**

The CMIP6 emission datasets analyzed during the current study are available at https://svn-ccsm-inputdata.cgd.ucar.edu/trunk/inputdata/atm/cam/chem/. The MERRA2 meteorological offline data are publicly available from https://rda.ucar.edu/datasets/ds313.3/.

**Competing interests**

The authors declare that they have no conflict of interest.

**Acknowledgements**

This work was supported by funding from the National Natural Science Foundation of China (under award nos. 42077196, 41821005) as well as the Newton Advanced Fellowship (NAFR2180103).

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
