# Peer review of "Influence of atmospheric in-cloud aqueous-phase chemistry on global simulation of SO2 in CESM2"

_Atmospheric Chemistry and Physics, 2021_

## Author Comment (AC1)

**Response to Referee #1**

We would like to thank Atmospheric Chemistry and Physics for giving us the opportunity to revise our manuscript. We thank the referees for their careful reading and thoughtful suggestions on the previous version of our manuscript. We have carefully addressed all of these valuable comments and revised our manuscript accordingly. Below are the point-to-point responses to the general and specific comments as well as our technical corrections.

*1. General*

*The paper points to the importance of aqueous iron chemistry in cloud droplets for the removal of anthropogenic $SO_2$ in the lower atmosphere in a chemistry climate model. There are other CCMs with similar aqueous chemistry and slightly different equations which are mostly cited in the introduction. It is, however, not clear with which boundary conditions the sensitivity studies concerning $Fe^{3+}$ and the pH are performed, also the definition of 'improved case' is fuzzy and has to be improved. Nevertheless the study is interesting and worth to be published after revision.*

**Response:** Thank you very much for your valuable comments. To make it more clear, we added a table describing all our model simulations used in this study in the supplement (see below), including chemical mechanisms, concentrations of soluble $Fe^{3+}$ ($[Fe^{3+}]$) and pH values used in the "Improved" case and all other cases associated to the sensitivity tests. The detailed reply to this comment about the boundary conditions is shown below (Response to *Comment* 3). At the same time, we also described this table briefly in the revised manuscript (Section 2.3, lines 221-222).

**Revisions:**

**Section 2.3, lines 221-222:**

"The detailed description of all the model simulations used in this study is summarized in Table S1 in the supplement."

**Table S1.** **Description of all model simulations.**

| No. | Case name | Location in the paper | Chemistry [a] | $[Fe^{3+}]$ (µM) | pH |
|-----|-----------|----------------------|---------------|------------------|-----|
| 1 | Original | Sect. 3.1, 3.2, 3.3 | a + b | | |
| 2 | Improved | Sect. 3.2, 3.3, 5.1 | a + c + d + e + f + g + h | 5 | calculated [b] |
| 3 | $HO_x$-chem | Sect. 4 | a + c + d + e | | calculated |

| No. | Name | Section | Mechanisms[a] | [Fe] | pH[b] |
|---|---|---|---|---|---|
| 4 | Fe-chem | Sect. 4 | a + c + d + f | 5 | calculated |
| 5 | N-chem | Sect. 4 | a + c + d + g | | calculated |
| 6 | Carbonate-chem | Sect. 4 | a + c + d + h | | calculated |
| 7 | Fe01 | Sect. 5.1 | a + c + d + e + f + g + h | 0.1 | calculated |
| 8 | Fe1 | Sect. 5.1 | a + c + d + e + f + g + h | 1 | calculated |
| 9 | Fe20 | Sect. 5.1 | a + c + d + e + f + g + h | 20 | calculated |
| 10 | Fe100 | Sect. 5.1 | a + c + d + e + f + g + h | 100 | calculated |
| 11 | pH3 | Sect. 5.2 | a + c + d + e + f + g + h | 5 | 3 |
| 12 | pH4 | Sect. 5.2 | a + c + d + e + f + g + h | 5 | 4 |
| 13 | pH5 | Sect. 5.2 | a + c + d + e + f + g + h | 5 | 5 |
| 14 | pH6 | Sect. 5.2 | a + c + d + e + f + g + h | 5 | 6 |
| 15 | $HO_x$-pH3 | Sect. 5.2 | a + c + d + e | | 3 |
| 16 | $HO_x$-pH4 | Sect. 5.2 | a + c + d + e | | 4 |
| 17 | $HO_x$-pH5 | Sect. 5.2 | a + c + d + e | | 5 |
| 18 | $HO_x$-pH6 | Sect. 5.2 | a + c + d + e | | 6 |
| 19 | Fe-pH3 | Sect. 5.2 | a + c + d + f | 5 | 3 |
| 20 | Fe-pH4 | Sect. 5.2 | a + c + d + f | 5 | 4 |
| 21 | Fe-pH5 | Sect. 5.2 | a + c + d + f | 5 | 5 |
| 22 | Fe-pH6 | Sect. 5.2 | a + c + d + f | 5 | 6 |
| 23 | N-pH3 | Sect. 5.2 | a + c + d + g | | 3 |
| 24 | N-pH4 | Sect. 5.2 | a + c + d + g | | 4 |
| 25 | N-pH5 | Sect. 5.2 | a + c + d + g | | 5 |
| 26 | N-pH6 | Sect. 5.2 | a + c + d + g | | 6 |
| 27 | Org-chem | Sect. 5.3 | a + c + d + i | | calculated |

[a] The chemical mechanisms corresponding to different letters are: a. the default MOZART-4 chemistry used in CAM4, b. default parameterized aqueous-phase oxidation reactions of $SO_2$ used in CAM4, c. gas-aqueous phase transfer equilibria in Table 1a, d. aqueous ionization equilibria, e. HOx-chemistry, f. Fe-chemistry, g. N-chemistry, and h. carbonate chemistry in Table 1b, and i. Organic chemistry in Tables S2a and S2b.

[b] The pH values in these simulations are calculated by gas-aqueous phase transfer equilibria in Table 1a and aqueous ionization equilibria in Table 1b.

*2. Specific*

*A statement that reactions in liquid aerosol are not included but only reactions in the gas phase and in cloud droplets, is at the beginning of section 2.2. It might be good to have that also earlier, even in the abstract.*

**Response:** Thank you again for this comment and we further revised the abstract and introduction accordingly, as shown below.

**Revisions:**

**Abstract, lines 13-20:**

"We replaced the default parameterized $SO_2$ aqueous-phase reactions with detailed $HO_x$-, Fe-, N- and carbonate chemistry **in cloud droplets** and performed a global simulation for 2014-2015. Compared with the observations, the results incorporating detailed **cloud** aqueous-phase chemistry greatly reduced $SO_2$ overestimation…This study emphasizes the importance of detailed **in-cloud** aqueous-phase chemistry for the oxidation of $SO_2$."

**Section 1, lines 115-120:**

"This study aims to examine the role played by detailed **in-cloud** aqueous-phase chemistry **(not including chemical reactions on the surfaces of wet aerosols)** on the capacity for oxidation of global $SO_2$ in the Community Earth System Model 2 (CESM2). We describe the CESM2 model, detailed **cloud** chemistry and observational data in Sect. 2. The evaluation of $SO_2$ simulations with or without **coupling detailed in-cloud** aqueous-phase chemistry is given in Sect. 3. The contributions of different **in-cloud** aqueous-phase chemical mechanisms to the simulation of $SO_2$ are analyzed in Sect. 4. The key factors that affect the capacity for $SO_2$ oxidation are discussed in Sect. 5. Finally, the main conclusions are drawn in Sect. 6."

3. *Section 2.1: What are the boundary conditions for aerosol? Is dust included or not? Is it related to the iron in the droplets? The provided links are not unique here since there are plenty of scenarios. More details please.*

**Response:** In this study, the component set we used is "FMOZ", which is the "CAM4 physics and MOZART-4 tropospheric chemistry with bulk aerosols" (https://www.cesm.ucar.edu/models/cesm2/config/compsets.html), just as said in Section 2.1. The default aerosol species simulated in this component set include sulfate, nitrate, ammonium, black carbon (BC), organic carbon (OC), secondary organic aerosol (SOA), dust and sea salt. Their emission, dry deposition and wet deposition processes are also guided by input files from CESM database (https://svn-ccsm-inputdata.cgd.ucar.edu/trunk/inputdata/atm/cam/chem/trop_mozart_aero/ ; https://svn-ccsm-inputdata.cgd.ucar.edu/trunk/inputdata/atm/cam/chem/emis/CMIP6_emissions_1750_2015_2deg/) and source codes of CESM2 (aero_model.F90, mo_drydep.F90 and wetdep.F90). These aerosols also use the same MERRA2 meteorological offline data for transport process.

Besides, as a global model, the boundary conditions of CESM2 are mainly related to the initial data file and spin-up process for the model. The initial data file consists of the initial values of all the physical variables and concentrations of all the chemical species, including all the aerosol species above. In this study, we used the initial data file "https://svn-ccsm-inputdata.cgd.ucar.edu/trunk/inputdata/atm/cam/inic/fv/cami-chem_1990-01-01_0.9x1.25_L30_c080724.nc" to set up the starting point of global concentrations, and then run a whole year (2014) for spin-up. Other settings for the simulations are shown in Table S1.

**All mentioned above** (emission, deposition, transport and background concentrations with their detailed links) **are the boundary conditions for aerosol** including **dust**. As for dust, there is also a boundary dataset for soil erodibility factors (https://svn-ccsm-inputdata.cgd.ucar.edu/trunk/inputdata/atm/cam/inic/fv/cami-chem_1990-01-01_0.9x1.25_L30_c080724.nc). However, the soluble Fe in the cloud droplets is not coupled with dust. For one reason, the real-time simulation of soluble Fe from dust involves multiple processes including calculating the proportion of elemental Fe in the dust, the solubility of total Fe and the liquid water content of clouds, all of which involve complex processes and subject to large uncertainties. In addition, the emission sources of dust in CESM2 are relatively incomplete, covered only natural sources (e.g., deserts). Anthropogenic emissions from steel, power and construction industries, especially in China, are not included. Therefore, it is difficult to precisely simulate soluble Fe from dust directly. Therefore, instead of calculating the soluble Fe from dust, we prescribed the soluble Fe concentration in cloud, which could more directly quantify the sensitivity of $SO_2$ oxidation to different levels of soluble Fe.

To respond this comment, we added the following sentences in Sections 2.1, 2.3 and 5.3 of the revised manuscript.

**Revisions:**

Section 2.1, lines 123-130:

"The Community Earth System Model 2 (CESM2, v2.1.1), developed by the National Center for Atmospheric Research (NCAR, https://www.cesm.ucar.edu/models/cesm2/, last access: 16 December 2020) is used in this study (Emmons et al., 2020; Danabasoglu et al., 2020), configured with the Community Atmosphere Model version 4.0 (CAM4). The coupled chemistry in CAM4 is primarily based on the Model for Ozone and Related chemical Tracers, version 4 (MOZART-4), including 85 gas-phase

species with bulk aerosols and detailed tropospheric chemistry with 196 gas-phase reactions (Emmons et al., 2010; Lamarque et al., 2012). **The default aerosol species simulated in this component set include sulfate, nitrate, ammonium, black carbon (BC), organic carbon (OC), secondary organic aerosol (SOA), dust and sea salt.** In this study, we develop a detailed aqueous-phase chemistry module for $SO_2$ oxidation fully coupled in the MOZART-4 chemistry."

**Section 2.1, lines 136-143:**

"All the emission inventories needed for MOZART-4 chemistry are obtained from the CESM database (https://svn-ccsm-inputdata.cgd.ucar.edu/trunk/inputdata/atm/cam/chem/CMIP6_emissions_1750_2015, last access: 31 December 2020 ), which was developed for the CMIP6 projects (Feng et al., 2020). The inventories have been updated to 2015, which is the year of the simulation in this study. **Meanwhile, the emission, dry deposition and wet deposition processes of aerosol species are also guided by input files from CESM database (https://svn-ccsm-inputdata.cgd.ucar.edu/trunk/inputdata/atm/cam/chem/trop_mozart_aero/ ; https://svn-ccsm-inputdata.cgd.ucar.edu/trunk/inputdata/atm/cam/chem/emis/CMIP6_emissions_1750_2015_2deg/) and the source codes of CESM2 (aero_model.F90, mo_drydep.F90 and wetdep.F90).**"

**Section 2.3, lines 223-228:**

"Finally, all the simulations are running for a 2-year period from $1^{st}$ January 2014 to $31^{st}$ December 2015. The first year is used for model spin-up. **In this study, we used "https://svn-ccsm-inputdata.cgd.ucar.edu/trunk/inputdata/input/atm/cam/inic/fv/cami-chem_1990-01-01_0.9x1.25_L30_c080724.nc" as the initial data and boundary conditions to provide the initial values of all the physical variables and concentrations of all the chemical species.** The output of the simulation is in the form of a daily mean and is then converted to a monthly or seasonal mean for research needs."

**Section 5.3, lines 497-498:**

"Therefore, the concentration of soluble Fe can be calculated in a range from less than $10^{-3}$ μM to $10^3$ μM, involving great uncertainties. **It is also the reason why the dust aerosol is simulated but not coupled with soluble Fe in this study.**"

*4. Table 1: Are the chemical equations complete? I wonder about missing HCHO and CH₃CHO and sometimes different products compared to other models. Is HONO in the text and HNO₂ in the table the same? If yes please decide for one notation.*

**Response:** All the reactions in Tables 1a and 1b are summarized from 17 references in total (see Tables 1a and 1b). Considering that the main object of this study is $SO_2$ and its oxidation, we focus only on the reactions related to $SO_2$ oxidation. Therefore, although Table 1 may not cover all the aqueous-phase reactions in the atmosphere, it is still a relatively comprehensive summary for in-cloud aqueous-phase oxidation of $SO_2$.

For HCHO and $CH_3CHO$, we didn't add relevant aqueous-phase reactions, because this study mainly focuses on $SO_2$ rather than organic compounds. Certainly, there are plenty of gas-phase reactions of HCHO/$CH_3CHO$ included in the default MOZART-4 chemistry, such as:

$CH_3OH + OH \rightarrow HO_2 + HCHO$

$CH_3O_2 + NO \rightarrow HCHO + NO_2 + HO_2$

$HCHO + NO_3 \rightarrow CO + HO_2 + HNO_3$

$HCHO + OH \rightarrow CO + H_2O + HO_2$

$CH_3CHO + OH \rightarrow CH_3CO_3 + H_2O$

$CH_3CHO + NO_3 \rightarrow CH_3CO_3 + HNO_3$

$C_2H_5OH + OH \rightarrow HO_2 + CH_3CHO$

To address this comment more clearly, we did add more aqueous-phase organic reactions in the simulation (as shown in Tables S2a and S2b in the supplement). However, we find that the addition of aqueous-phase organic chemistry did not have a significant effect on $SO_2$ simulation. We described this new simulation in Section 5.3 of the revised manuscript.

Finally, according to the definition of HONO in *Atmospheric Chemistry and Physics* (2nd edition, 2006, Chapter 6.7, p231), "Nitrous acid (i.e., $HNO_2$), HONO, is formed by a heterogeneous reaction involving $NO_2$ and $H_2O$." (Seinfeld and Pandis, 2006). Therefore, HONO and $HNO_2$ are the same in our chemical mechanisms. They're just different expressions of the same kind of molecule. This time we unified these two expressions as $HNO_2$ (in lines 73 and 87).

**Revisions:**

**Section 1, lines 73-74:**

"Moreover, these studies have indicated that > 95% of $NO_2$ converts to **$HNO_2$**/$NO_2^-$ by hitting the surface of $NaHSO_3$ aqueous microjets to promote the aqueous-phase oxidation of $SO_2$."

**Section 1, lines 85-87:**

"Some laboratory studies have focused on the detailed mechanism, such as the radical processes involved in different pathways of the Fenton reaction and the conversion of $NO_2$ to **$HNO_2$** to oxidize $SO_2$."

5. *Section 2.2, after line 186 and section 2.3: An initial setting of [$Fe^{3+}$] cannot explain how the different scenarios and the spatial distributions are constrained. More information please. Do you mean every time step at cloud formation, if meteorological conditions are favorable for the selected cloud types?*

**Response:** Take the "Improved case" for example. At t = 0 of each timestep, all the droplets of large-scale liquid stratiform clouds (selected cloud types) are formed according to the cloud-related variables such as LWC, $F_{cld}$ and r. At the same time, a given **initial** concentration of soluble $Fe^{3+}$ (5 µM) is allocated into each droplet. It should be noted that what we control is only the initial concentration at the very beginning of each timestep. Then these $Fe^{3+}$ will participate into aqueous-phase reactions and the concentrations and proportions of soluble Fe in different valences will change all the time during the timestep. Therefore, although the initial concentration of soluble Fe at each timestep is uniformly distributed, there are still different spatial distributions of all kinds of soluble Fe species that really involved in the reactions. At the same time, these spatial distributions are also directly related to the cloud variables in different regions, which could also change the distributions.

We also conducted more sensitivity tests with different levels of soluble $Fe^{3+}$ in Section 5.1 to further understand the $SO_2$ oxidation capacity in response to a broader range of soluble Fe concentrations. These additional four levels of soluble $Fe^{3+}$ are chosen based on a summary from multiple previous studies (shown in lines 389-393) and could basically cover the range of soluble $Fe^{3+}$ levels on land. Furthermore, due to the relatively short lifetimes of $SO_2$ and soluble Fe (regional pollutants), these sensitivity tests could generally match their variability in different regions. For instance, the "initial [$Fe^{3+}$] = 20 or 100 µM" scenarios fit some industrialized regions in China well. We added more descriptions in Sections 2.3 and 5.1 in the revised manuscript (see below).

**Revisions:**

"The timestep used in this study is the default 30 minutes in CESM2. **In the Improved case,** the lifetime of clouds (i.e., the time between the formation and evaporation of clouds) is set equal to the timestep. At t = 0 of each timestep, all the cloud droplets are assumed to be instantaneously and simultaneously formed **according to the cloud-related variables such as LWC, $F_{cld}$ and r**, and all the water-soluble species (listed in Table 1a) are dissolved into the cloud droplets according to the effective Henry's law constants. The pH value of each grid cell is calculated by the ionization equilibria of ionizable species (listed in Table 1b) and the dissociation of CCN (including sulfate, nitrate and ammonium), assuming that equilibrium and electroneutrality are continuously maintained. **Such pH values can directly influence the formation of aqueous-phase sulfate and nitrate of this timestep. At the same time, a given initial concentration of soluble $Fe^{3+}$ (5 μM) is allocated into each cloud droplet.** When t = 30 minutes, all the cloud droplets are assumed to instantaneously evaporate. All the species remaining in the aqueous phase are transferred directly back to the gas phase. Low-volatility species such as ammonium, sulfate and nitrate are released directly back to the atmosphere as inorganic aerosols. **Meanwhile, the newly formed sulfate and nitrate will further influence the ionization equilibria and the calculation of pH values in the next step, thus forming a fully-coupled feedback system between pH values and concentrations of sulfate and nitrate.**"

"Therefore, in this study, four other levels of **initial** $[Fe^{3+}]$ (0.1, 1, 20 and 100 μM) are tested **with the whole in-cloud aqueous-phase reactions** to evaluate the influence of soluble Fe concentration on the capacity for $SO_2$ oxidation. **The processing $[Fe^{3+}]$ in these sensitivity cases is identical to the Improved case except the differences in $Fe^{3+}$ concentrations.** All the levels of $[Fe^{3+}]$ are based on the reported values above. **See Table S1 for details.**"

6. *Section 3: In the introduction reactions on mineral dust are mentioned, but not later in connection with the mismatch between model and observations in China, please improve. It might be also useful to include an extra line in Fig 4c with high Fe (e.g. from dust in droplets). Or is this automatically included in China?*

**Response:** Thank you again for your careful reading. As mentioned in our Responses to *Comments 2 and 3*, reactions on the surfaces of dust aerosols are not included in this study, and it is not the purpose of this study to specially simulate the distribution of mineral dust as well as soluble $Fe^{3+}$. Admittedly, surface reactions on mineral dust and wet aerosols have been pointed out to be also very important in more and more studies in recent years, especially in China. And we will also consider taking these kinds of reactions into our future work.

For the improvement of simulation in China, we indeed drew the simulation line with high Fe in Figure 6c.

7. *Section 3: Table 2, Figures 4, 6 and 7: The different units are very distracting. Better decide for one common unit and provide the (approximate) conversion factors in the caption of Table 2 or the text.*

**Response:** Actually, the units of observational data from these four monitoring networks themselves are different. And we indeed had intended to unify the units of these data, but the database platforms did not provide corresponding observed information on temperature and pressure, which differ a lot in different regions and at different times. Therefore, there are no specific and fixed unit conversion factors and we had to let the units of simulated data follow those of observational data.

8. *Section 4, line 344: I suppose these studies are performed with and without the corresponding equations of Table 1. Please mention that clearly if that is the case or explain how you proceeded.*

**Response:** All the simulation in Section 4 are performed with the corresponding equations in Table 1 and see Table 1 and Table S1 for details. Considering that the $HO_x$ chemistry involves most of the critical radicals in aqueous-phase chemistry, the other three cases of Fe, N and carbonate chemistry also include the $HO_x$ chemistry. Then the contributions of Fe, N and carbonate chemistry are calculated by the differences between the results of their corresponding cases and that of the $HO_x$-chem case. To better explain the simulation of this section, we added the explanation above into Section 4 of the revised manuscript.

**Revisions:**

Section 4, lines 357-362:

"On the basis of above analysis of the overall detailed aqueous-phase chemistry, it is necessary to discuss the contributions of different aqueous-phase chemical mechanisms in detail. **Cases for four different mechanisms are performed with the corresponding reactions in Table 1. See Table S1 for details about the configuration of individual cases. Given the fact that the $HO_x$ chemistry involves most of the critical radicals in aqueous-phase chemistry, the cases of Fe, N and carbonate chemistry also include the $HO_x$ chemistry. The individual contribution of Fe, N or carbonate chemistry is compared with the $HO_x$-chem alone case.**"

9.  *Section 5.1, line 370 (and earlier): it not quite clear how [$Fe^{3+}$] at different values is superimposed to the variable values at different locations. It should be better explained already in Section 2 what the "improved case" is with this respect.*

10. *Line 376: added to what (and where)?*

**Response:** We have responded these questions in our Responses to *Comments 3 and 5*. At the beginning of each timestep, a given initial concentration of soluble $Fe^{3+}$ (0.1, 1, 5 (used in the "Improved" case), 20 or 100 μM) is allocated into each cloud droplet. Then these $Fe^{3+}$ will participate into aqueous-phase reactions and change all the time during the timestep. The detailed settings of the Improved case and other sensitivity cases with different concentrations of soluble $Fe^{3+}$ are shown in Table S1.

Besides, these $Fe^{3+}$ are added into the whole in-cloud aqueous-phase reactions of each cloud droplets at the beginning of each timestep. This has been added in lines 395-396 of Section 5.1 (as shown above). Other revisions to the manuscript related to these two comments are also shown in our responses to *Comments 3 and 5*.

11. *Section 5.2: Please more details how the pH is entered into the simulations. Is it only via the directly dependent equations or for example via constraining of [$H^+$] using Eqns. 33 to 58 of Table 1 in the droplets? Feedbacks with sulfate and nitrate as indicated in section 2.3? Please improve text here.*

**Response:** Firstly, in the Improved case and cases in Sections 4 and 5.1, the pH values are calculated by [$H^+$] using Reactions 1 to 58 of Table 1 in the cloud droplets at the beginning of each timestep, because

the gas-aqueous transfer process and ionization process are in a whole equilibrium system. However, in the sensitivity cases for pH values in Section 5.2, pH is prescribed as a fixed value during the timesteps. Such a setting is more suitable for quantitatively studying the effects of pH value on the capacity for $SO_2$ oxidation. These settings are also summarized in Table S1.

For the former kind of $[H^+]$ and pH value in each grid, they are calculated by the initial concentrations of all ionizable aqueous-phase species (including sulfate and nitrate) and their Henry's law and ionization constants in Table 1 at the beginning of each timestep, just as said in lines 207-213 of the manuscript. Such pH values can directly influence the formation of aqueous-phase sulfate and nitrate of this timestep. After a timestep of aqueous-phase reactions, the newly formed sulfate and nitrate will further influence the ionization equilibria and the calculation of pH values in the next step, thus forming a fully-coupled feedback system between pH values and concentrations of sulfate and nitrate. As for the effects of pH value and even the whole in-cloud aqueous-phase chemistry on sulfate and nitrate, it is not the scope of this work, but the main target of our next stage of study.

To improve the description of pH value calculation, we added above interpretations into Sections 2.3 and 5.2 in the revised manuscript (see below).

**Revisions:**

Section 5.2, lines 427-430:

"Therefore, it is necessary to discuss the influence of the variation of pH value on the capacity for $SO_2$ oxidation. In this study, **there are four sets of pH values (i.e., 3, 4, 5 and 6) prescribed in the following sensitivity tests (Table S1).** All the pH values are referenced from previous studies (Herrmann et al., 2000; Matthijsen et al., 1995; Shao et al., 2019; Guo et al., 2017; Cheng et al., 2016)."

*12. What is the difference between Fig. S7 (former Fig. 7 in first draft) and Fig. S10? You may skip one because there appears to be almost no difference (except a slightly different caption and differences in Africa for JJA and SON for pH 6). Or explain better in text (section 5 and 2) and the captions.*

**Response:** Thank you again for this suggestion. Fig. S7 shows the differences in $SO_2$ mixing ratios **between the sensitivity cases with different prescribed pH values and the Improved case where pH**

**value is calculated online (i.e., case 11~14 – case 2**). Cases 11-14 use the same chemical mechanism as the Improved case except for different pH values. While Fig. S10 shows the differences in $SO_2$ mixing ratios at different pH values **with the Fe-chemistry only (i.e., case 19~22 – case 4).** These descriptions have been added into the figure captions. At the same time, the detailed information of these sensitivity tests is also added in Table S1 and Section 5.2 of the revised manuscript (lines 428-429). The reason why these two figures are similar is that the contribution of Fe-chemistry plays a dominant role and thus omitting the N-chemistry leads to a trivial change.

*13. Technical corrections*

*Please provide maxima and minima or the range of the color bars in the figures (also in supplement).*

**Response:** Thanks for this suggestion. For Figure 2 and S2, we added the minimum "0" in the color bars. For other maps, the ends of color bars just represent "more than" or "less than" a value. Therefore, there are no fixed or clear minimum or maximum for these maps. In fact, many other published papers in ACP also have a similar style in color bars, e.g., (Tao et al., 2017; Liu et al., 2018; Im et al., 2018; Wei et al., 2019; Huang et al., 2019; Xu et al., 2019; Buchard et al., 2014; Hedegaard et al., 2008; Pozzer et al., 2012).

*14. Include blanks after ';' in citations.*

**Response:** Thank you again for pointing out this mistake. We have updated the Endnote style template of Copernicus Publications Copy and added the blanks after ';' in citations successfully. Please see the revised manuscript.

*15. Line 356, 393: Don't use 'trend' here, wrong word.*

**Response:** Thanks. We modified these two expressions in the revised manuscript as follows.

**Revisions:**

Section 4, line 374:

In regard to carbonate chemistry, however, it is difficult to see consistent **change in either spatial or temporal SO$_2$ distribution**.

Section 5.1, line 414:

Such an **effect** on the capacity for SO$_2$ oxidation by [Fe$^{3+}$] chemistry can also be seen in Fig. S6.

16. *In the references often page numbers are missing. The doi is provided with 'DOI', 'doi' or nothing before, please use consistent notation.*

**Response:** Thanks again. We added the missing page numbers of references and unified the expression of DOI (chose "nothing before"). Please see the revised manuscript.

**References**

[revised manuscript text omitted]

---

## Author Comment (AC2)

**Response to Referee #2**

We would like to thank Atmospheric Chemistry and Physics for giving us the opportunity to revise our manuscript. We thank the referees for their careful reading and thoughtful suggestions on the previous version of our manuscript. We have carefully addressed all of these valuable comments and revised our manuscript accordingly. Below are the point-to-point responses to the general and specific comments as well as our minor corrections.

**1. General comment**

This manuscript concerns with the global impact of aqueous-phase chemistry on the simulation of atmospheric SO2 with the CESM2 model. They apply a framework for integrating this chemistry that is similar to the one in use in other models like GEOS-Chem. This approach does not foresee the online calculation of pH. Additionally, not having soluble Fe emissions the authors go on with sensitivity simulations with plausible, at least regionally, of pH and  $[Fe^{3+}]$  applied globally. Then, an attempt of separating the effects of different parts of the mechanism on SO2 is done but limited by the choice of the reaction categories to exclude at once (more to it below). In general, the manuscript is written well and with a good and clear structure. The topic and the results are interesting. However, I have identified a few points to be addressed/clarified.

**Response:** Thank you very much for your valuable comments. As we introduced in lines 210-213 of Section 2.3, the pH values in the Improved case and other cases in Sections 4 and 5.1 are all calculated online by the equilibrium reactions in Tables 1a and 1b. Only the pH values in Section 5.2 are prescribed as a fixed value. See Table S1 for details.

On the other hand, the separation of effects did not directly exclude all the other reaction categories at once. We still retain the basic equilibria and radical reactions in  $HO_x$ -chemistry in every case (Please see below and Table S1 for details).

**2. Comments**

In Table 1b most of the references for the reaction kinetics are given to three collections/mechanisms published earlier. Please cite alongside the primary literature for each reaction and not just the modelling

studies that collected sets of reactions. Moreover, it is not explained/justified why the authors blends these (secondary) sources with others (primary) to obtain their own chemical mechanism in CESM.

Although section 2.2 contains the reaction tables that well done (apart from the secondary references), in the text little is written for describing in words the salient features of the mechanism used here.

I understand the authors wrote a comprehensive introduction on the SO2-relevant known chemistry. Howeover, the latest and most reliable data is not necessarily reflected in their mechanism. For instance, the oxidation of S(IV) by NO2 (also mentioned in the text). However, in Table 1b one finds only reaction 199 for  $HSO_3^-$  and nothing for  $SO_4^-$ . Is the latter neglected because only important at high pH values that are seldom represented in model simulations?

Furthermore, although Cheng et al., 2016 is cited in the introduction, the low rate constant by Lee and Schwarzt 1982 (by the way the reference is incomplete, please add at least the url https://www.osti.gov/biblio/6567096) is used. This is a 2nd-order reaction rate constant while reaction 199 is obviously 3rd-order. The former is about one order of magnitude larger as reported by a few later studies with the last one being Spindler et al. 2000, DOI: 10.1016/S1352-2310(03)00209-7. These issues I have make me doubt about the statement on line 18 in the abstract about the minor role of N-chemistry for SO2 oxidation.

**Response:** Thank you for your valuable comments. For the references, in this version of our manuscript we replaced all the secondary sources with primary sources (119 more references in all), as shown in Tables 1a and 1b in the revised manuscript. Meanwhile, the general description of these mechanisms is in lines 155-162 and the illustration of parameters is in lines 180-184 in Section 2.2.

Considering that the main focus of this study is SO2, we tried our best to summarize the reactions related to SO2 oxidation, although Table 1 may not include all the aqueous-phase reactions in the atmosphere. As for the oxidation of S(IV) by NO2, just as you mentioned, this is partially because the pH values of the cloud droplets are mostly in the range of 3-5.5, but the pKa of H2SO3 are 1.76 and 7.21, respectively (Herrmann et al., 2000). Therefore, the main form of S (IV) in cloud droplets is HSO3- and we neglected the latter in model simulations.

Next, we supplemented the URL of Lee and Schwartz (1982) into the footnote of Table 1b. In Lee and Schwartz (1982), they firstly determined the stoichiometry of the reaction as "2 NO2 + SO2" according to their experiments. Then they tried to convert the  $3^{rd}$ -order rate constant to  $2^{nd}$ -order (pseudo-second-order rate constant). Finally they set the  $2^{nd}$ -order reaction rate constant as  $2.0 \times 10^{6} \text{ M}^{-1} \text{ s}^{-1}$ . This rate constant,

for reaction rate interpreted according to an overall second-order rate expression ( $R = k(2)[S(IV)][NO_2]$ ), was  $2 \times 10^6 \text{ M}^{-1} \text{ s}^{-1}$  at both pH 6.40 and 5.80. Meanwhile, many other studies also referred to this work and used  $2 \times 10^6 \text{ M}^{-1} \text{ s}^{-1}$  as the 2nd-order reaction rate constant, including the references cited in next comment (Seinfeld and Pandis, 2006; Shao et al., 2019; Song et al., 2021).

Compared to Spindler et al. (2003), the reaction used in this study is  $NO_2 + SO_2 \rightarrow NO_2^- + SO_3^-$ . It is just an incomplete reaction and still need one more  $NO_2$  to oxidize  $SO_3^-$  (S(V)) to  $SO_4^{2-}(S(VI))$ . While Reaction 199 in our study is an integrated expression of these two steps, so they do not conflict to (Spindler et al., 2003).

3. It is remarkable that the mechanism the authors put together completely lacks the chemistry of methanol, methyl hydroperoxide and formaldehyde. The latter, in its hydrated form, combines with HSO3- to produce HMS which reacts quickly with OH to yield SO5-. Recent work on the importance of such chemistry are DOI: 10.1029/2020JD032706 and DOI: 0.5194/acp-21-457-2021. Why only reactions of organic acids with sulfur are considered? Why has this chemistry been neglected? It is well known and acknowledged as a significant source of O2- and thus OH. I think this chemistry needs to be considered especially in a study about the importance of SO2 aqueous-phase sinks.

**Response:** Thanks a lot for this important suggestion. We followed the referee's suggestion and incorporated additional 10 aqueous-phase organic species and 60 related reactions, including the reactions relevant to CH3OH, HCHO, CH3OOH and HMS-, as shown in Tables S2a and S2b. We conducted additional two simulations for testing the contribution from this organic chemistry (i.e., (1) "Improved case + Org-chem" and (2) "HOx-chem case + Org-chem"). The differences between case (1) and the "Improved case" are shown in Fig. R1 and the differences between case (2) and the "HOx-chem case" are shown in Fig. R2. Both figures indicate the influence from this organic chemistry is typically less than 5%. More specifically, the concentration of SO2 over China and mixing ratio over the United States only decreased by 1.2% and 0.2%, respectively (as shown in Fig. R3). Consequently, adding these organic species and reactions did not improve the simulation significantly. We added the following discussion in Section 5.3 of the revised manuscript.